# Human intracranial recordings link suppressed transients rather than 'filling-in' to perceptual continuity across blinks

Tal Golan[1], Ido Davidesco[2], Meir Meshulam[3], David M Groppe[4,5], Pierre Mégevand[4,5], Erin M Yeagle[4,5], Matthew S Goldfinger[4,5], Michal Harel[3], Lucia Melloni[6,7], Charles E Schroeder[8,9], Leon Y Deouell[1,10], Ashesh D Mehta[4,5†], Rafael Malach[3*†]

[1]Edmond and Lily Safra Center for Brain Sciences, The Hebrew University of Jerusalem, Jerusalem, Israel; [2]Department of Psychology, New York University, New York, United States; [3]Department of Neurobiology, Weizmann Institute of Science, Rehovot, Israel; [4]Department of Neurosurgery, Hofstra Northwell School of Medicine, Manhasset, United States; [5]The Feinstein Institute for Medical Research, Manhasset, United States; [6]Department of Neurophysiology, Max Planck Institute for Brain Research, Frankfurt am Main, Germany; [7]NYU Comprehensive Epilepsy Center, Department of Neurology, School of Medicine, New York University, New York, United States; [8]Department of Neurosurgery, Columbia University College of Physicians and Surgeons, New York, United States; [9]Cognitive Neuroscience and Schizophrenia Program, Nathan Kline Institute, Orangeburg, United States; [10]Department of Psychology, The Hebrew University of Jerusalem, Jerusalem, Israel

**\*For correspondence:** rafi. malach@gmail.com

[†]These authors contributed equally to this work

**Competing interests:** The authors declare that no competing interests exist.

**Abstract** We hardly notice our eye blinks, yet an externally generated retinal interruption of a similar duration is perceptually salient. We examined the neural correlates of this perceptual distinction using intracranially measured ECoG signals from the human visual cortex in 14 patients. In early visual areas (V1 and V2), the disappearance of the stimulus due to either invisible blinks or salient blank video frames ('gaps') led to a similar drop in activity level, followed by a positive overshoot beyond baseline, triggered by stimulus reappearance. Ascending the visual hierarchy, the reappearance-related overshoot gradually subsided for blinks but not for gaps. By contrast, the disappearance-related drop did not follow the perceptual distinction – it was actually slightly more pronounced for blinks than for gaps. These findings suggest that blinks' limited visibility compared with gaps is correlated with suppression of blink-related visual activity transients, rather than with "filling-in" of the occluded content during blinks.

## Introduction

The perceived continuity of the visual input despite its frequent interruptions by spontaneous eye blinks is a ubiquitous and powerful dissociation between sensation and perception. As such, it offers an ecological test of candidate neural correlates of visual awareness.

The perceptual omission of one's own eye blinks cannot be explained by the blinks' apparent briefness: blinks occlude the pupil for a considerable amount of time, typically 100–150 ms (*Riggs et al., 1981*). External darkenings of such duration have been shown to cause a robust percept (*Riggs et al., 1981*). Strikingly, since blinks normally occur at least 1000 times an hour (*Cruz et al., 2011*), about three or four percent of our waking hours are unknowingly spent with our

**eLife digest** The average person blinks once every few seconds, each time shutting off their view of the world for about a tenth of a second. Nevertheless, we rarely notice a blink. By contrast, we readily notice a single blank frame in a movie, even if the frame lasts far less than a blink. The fact that we do not usually notice our spontaneous blinks is a striking example of the discrepancy between the images we perceive versus the information that enters our eyes.

This dissociation between the information that the eyes receive and what we perceive raises a number of questions. First, which brain areas represent the actual information from the eyes, and at what point do brain areas start to represent our subjective perception instead? Second, how does the brain "stabilize" our perception of vision despite the frequent interruptions that occur whenever we blink? In short, does the brain "fill in" the missing images or "edit out" the gaps?

To answer these questions, Golan et al. turned to human patients who were undergoing a surgical procedure related to the treatment of epilepsy. In the course of such procedures, and strictly for diagnosis purposes, electrodes are temporarily placed directly on the surface of the brain – the cortex – making it possible to monitor the activity of individual cortical areas. Towards the back of the brain, where cortical processing of visual signals begins, neurons responded in a way that was consistent with the physical information the eye actually received rather than the perception of vision. Thus, neurons showed the same responses to easily seen blank frames in a movie as to unnoticeable blinks. However, as the signals streamed forward to down-stream brain regions involved in vision, neurons in successive areas were increasingly likely to distinguish between the perceptually visible blank frames versus the invisible blinks.

Unexpectedly, Golan et al. found no evidence that the brain fills in the missing picture during blinks. Instead, it seems that the brain generates a continuous perception by actively "deleting" the brief neural signals that are turned on when our visual input has been shut off. The brain only does this for blinks but not for artificial interruptions – such as blank movie frames – which explains why we notice the latter but not the former.

A future challenge will be to isolate the pathway that leads from the brain regions that generate blinks to the regions that deal with vision, and that enables us to tell blinks from blanks.

eyes closed. While spontaneous blinks are invisible, voluntary blinks are not. However, their perception is also not veridical: they are experienced as shorter and less dim than physically comparable artificial darkenings (*Riggs et al., 1981*).

Human psychophysical experiments have found evidence for a decrease in visual sensitivity during blinks even when the optical-retinal impact of blinks was neutralized (*Volkmann et al., 1980*). This effect could be mediated by an extra-retinal suppression of the neural response in early visual cortices during blinks, as indicated by feline V1 single unit recordings (*Buisseret and Maffei, 1983*) and human fMRI (*Bristow et al., 2005b*).

Whereas the evidence for blink-related suppression of early visual activity may explain why visual sensation is reduced during blinks, it does not readily explain the perceived continuity of the visual scene across blinks. Consider a reduction of retinal input driven by an external origin, such as when someone briefly turns off the lights. Such a reduction will diminish low-level visual activity as well, yet, unlike blinks, this external reduction is clearly visible. Hence, it seems that a neural basis for the special perceptual status of blinks requires a representation that is both unperturbed by blinks and yet still sensitive to perceived external darkenings. Furthermore, if one assumes that perceived continuity relies on a read-out of an ongoing representation of the visual scene, this entails an active 'filling-in' of this visual representation during blinks but not during external darkenings (*Billock, 1997*). Operationally, a filling-in mechanism would be reflected in continuous neural activity across blinks but not across gaps despite the decrease in retinal input common to both. This hypothetical effect is in an opposite direction to the previously reported neuronal suppression.

Critically, testing these predictions using human neuroimaging requires sufficient spatiotemporal resolution to distinguish between activity changes that occur prior, during and following the blink event across different visual regions. In particular, the sluggish BOLD-fMRI signal (used in previous

related studies, e.g., *Bristow et al., 2005a*) can register only a temporal average of the total blink-related changes, potentially summing over antagonistic positive and negative components.

Here, we overcame this limitation by examining the effect of spontaneous and voluntary eye blinks and brief external image disappearances ('gaps') on visual representations in human patients undergoing intra-cranial electrocorticographic evaluation for intractable epilepsy. This approach allowed us to test how these brief events interact with object-related human visual responses on a millisecond/millimeter-scale across multiple visual regions recorded simultaneously.

Following previous findings in paradigms unrelated to blinks (*Fisch et al., 2009*; *Moutard et al., 2015*), we hypothesized that the high-frequency broadband power envelope (HFB) response of the local field potential in human high-level ventral visual cortex would reflect the perceptual distinction between extrinsic disappearance of the stimuli and their disappearance due to spontaneous eye blinks. Furthermore, we sought to directly test the intuitive yet untested conjecture that missing content due to blinks is actively filled-in by sustained neuronal activity, whereas perceived external stimulus disappearances leave 'dips' in neural responses.

In brief, we found a posterior-anterior gradient in blink versus gap representations. In early visual cortex, these perceptually-distinct events elicited similar HFB responses, whereas in higher-level visual cortex, the termination of gaps elicited considerable overshoot beyond baseline levels, an effect absent in spontaneous blinks. Intriguingly, both low and high-level cortical sites failed to exhibit a differential filling-in for blinks compared to gaps, suggesting that the perceived continuity of the visual scene might not depend on a continuous neural representation in category-selective visual areas but on the lack of representation of discontinuities, that is, stimulus disappearances and reappearances.

## Results

Fourteen patients undergoing electrocorticographic evaluation for intractable epilepsy participated in the study (see *Table 1*). The patients viewed consecutively presented grayscale photographs of faces and non-face images from several categories (houses, tools, abstract patterns and animals). The patients clicked the mouse button each time they detected an animal image (mean hit-rate = 86.1%, mean false-alarm rate = 3.4%). Target (animal) trials were excluded from further analysis. The images were presented at a pace of one image per second with no inter-stimulus blanks (see *Figure 1* and Materials and methods). The patients were concurrently monitored for eye blinks by a video eye tracker and an electrooculogram (EOG). Once every ten images, a gray screen was displayed for three seconds, serving as a baseline and partitioning the trials into ten-trial long blocks. Critically, in some of the trials, the displayed stimuli were interrupted by either spontaneously produced eye blinks, periodic (~1 Hz) voluntary eye blinks whose production was cued at the beginning of some of the blocks, or 'gaps', produced by replacing the stimulus with a black or gray screen for variable latencies and durations, aimed at evaluating the purely retinal impact of blinks. Individual latencies and durations of the three kinds of interrupting events for each patient are presented in *Figure 1—figure supplement 1*.

### Defining visually responsive electrodes

Whereas blinks may be correlated with neural responses also in non-visual regions (e.g., motor, somatosensory responses or even default mode network, see *Nakano et al., 2013*), in the present study we were concerned exclusively with the modulation of visual representations by blinks. Therefore, we first tested which electrodes reliably responded to the presented stimuli (e.g. faces) themselves. Since we were interested in local field potential correlates of average spiking rate, all of our analyses were conducted on the high-frequency broadband power envelope (sampled between 70 and 150 Hz). A number of previous studies have shown the HFB signal to be a good index of the aggregate firing rate (*Mukamel et al., 2005*; *Ray et al., 2008*; *Rasch et al., 2008*; *Manning et al., 2009*; *Nir et al., 2007*). Following standard preprocessing and HFB computation, we tested the onset response (50–350 ms, uncontaminated by either gaps or blinks) to each electrode optimal object stimuli, compared with the inter-block gray blank baseline periods (*Figure 2*). In agreement with previous reports (*Noy et al., 2015a*), responsivity of a considerable effect size was found almost entirely within the anatomically defined visual cortex, showing only minimal responses in more anterior cortical regions.

Table 1. Patients' demographic, clinical and experimental details.

| Patient code* | Sex | Age | Seizure onset zone(s) | Voluntary blinks blocks | Black /gray gap control | Gradual / abrupt gap control | Total analyzed electrodes | Total visually responsive electrodes | Number of visually responsive electrodes in each ROI | | | | | | |
| | | | | | | | | | Retinotopic | | | | | High-level | |
| | | | | | | | | | V1 | V2 | V3 | V4 | VO | FC | N-FC |
|---|---|---|---|---|---|---|---|---|---|---|---|---|---|---|---|
| P20 | F | 30 | RH: Supramarginal Gyrus | ✓ | | | 103 | 6 | 0 | 0 | 2 | 0 | 0 | 1 | 2 |
| P25 | M | 45 | RH: Inferior Frontal Gyrus, Precentral S. | ✓ | | | 117 | 8 | 0 | 0 | 0 | 0 | 0 | 2 | 6 |
| P32 | M | 23 | RH: Superior Temporal Gyrus, Hippocampus | ✓ | | | 181 | 10 | 2 | 2 | 0 | 1 | 0 | 1 | 2 |
| P33 | F | 52 | LH: Hippocampus, Middle Entorhinal Cortex | ✓ | | | 83 | 8 | 0 | 0 | 0 | 0 | 0 | 4 | 1 |
| P36 | M | 24 | RH: Parahippocampal Gyrus, Temporal Pole | ** | ✓ | | 58 | 8 | 0 | 0 | 1 | 0 | 1 | 1 | 4 |
| P39 | M | 25 | RH: Hippocampus, Amygdala | ✓ | ✓ | | 128 | 18 | 3 | 2 | 2 | 1 | 2 | 3 | 4 |
| P44 | M | 30 | RH: Anterior Temporal Lobe | ✓ | ✓ | | 118 | 10 | 1 | 1 | 1 | 0 | 1 | 1 | 1 |
| P46 | M | 45 | RH: Hippocampus, Parahippocampal Gyrus | ✓ | ✓ | ✓ | 58 | 15 | 3 | 2 | 2 | 1 | 2 | 1 | 4 |
| P47 | F | 34 | LH: Anterior Temporal Lobe | ✓ | ✓ | ✓ | 142 | 7 | 0 | 0 | 0 | 1 | 1 | 3 | 2 |
| P50 | M | 27 | LH: Amygdala, Hippocampus, Parahippocampal Gyrus, Anterior Fusiform Gyrus, Post Central Gyrus | ✓ | ✓ | ✓ | 108 | 3 | 2 | 1 | 0 | 0 | 0 | 0 | 0 |
| P54 | M | 21 | RH: Medial Temporal, Middle Occipital Gyrus, Parieto-Occipital-Sulcus, Middle Temporal Gyrus, LH: Hippocampus | ✓ | ✓ | ✓ | 160 | 23 | 2 | 3 | 4 | 2 | 0 | 2 | 0 |
| P57 | M | 29 | RH: Amygdala | | ✓ | ✓ | 110 | 6 | 1 | 0 | 1 | 0 | 0 | 0 | 2 |
| P59 | M | 50 | RH: Parieto-Occipital Sulcus | | ✓ | ✓ | 94 | 13 | 3 | 0 | 0 | 0 | 1 | 0 | 2 |
| P62 | F | 44 | LH: Hippocampus, Anterior Cingulate Gyrus, Amygdala, Parahippocampal Gyrus | | ✓ | ✓ | 125 | 8 | 0 | 1 | 2 | 1 | 0 | 0 | 0 |

LH/RH – left/right hemisphere, VO – ventral-occipital, FC – face-selective electrodes, N-FC non-face selective high-level electrodes. * Patients' identities were coded by order of admission to surgery. Since not every admitted patient performed the experiment, the codes are not consecutive. ** Failed to follow the instruction to voluntary blink due to language barrier.

143 Electrodes out of a total of 1585 were found to show visual responses that were both significant (Wilcoxon rank-sum test, $p < 0.05$, Bonferroni-corrected within-patient across electrodes) and of a considerable effect size (Glass' $\Delta \geq 2$, see Materials and methods). These electrodes were qualified for subsequent analyses.

We chose face-preference as a model for ventral category-selectivity, motivated by its prevalent occurrence in previous recordings (*Privman et al., 2007*). Thus, responsive electrodes were assessed for face-selectivity by comparing the responses to face-stimuli with each non-face category. This analysis identified 19 electrodes (found in 10 of the 14 patients, see *Table 1*) that responded significantly more strongly to faces than to any other category (Wilcoxon rank sum test, $p < 0.05$ for all contrasts, uncorrected), located mostly in high-level ventral visual cortex.

## Disentangling the effect of gaps and blinks from the stimulus-driven visual response

In order to isolate the effect of interruptions (blinks and gaps) from the stimulus-driven responses, the response to the stimulus itself had to be accounted for first. We have approached this using time-domain deconvolution. This is similar to the way signals are unmixed in the analysis of fast event-related fMRI (*Burock and Dale, 2000*). This procedure was implemented as a multiple linear regression of the observed timecourse with a set of finite impulse response bases (see *Figure 3* and

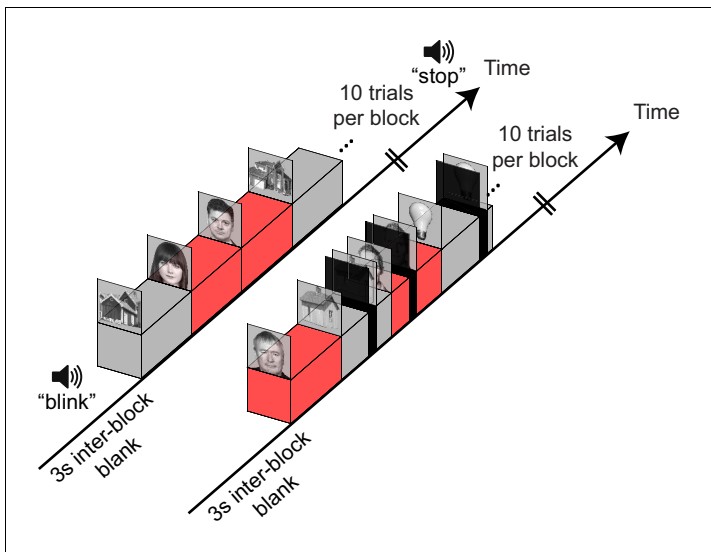

**Figure 1.** Experimental design. Two blocks of the experimental task are illustrated. All stimuli are presented for 1 s each. The block on the left is preceded with an auditory instruction to execute voluntary blinks about once a second. The block on the right is interleaved with experimenter-induced black gaps (at 350, 550 or 750 ms following trial-onset). See *Figure 1—figure supplement 1* for individual latency and duration distributions of gaps, voluntary blinks and spontaneous blinks.
Face photographs: Owen Lucas. Available on Flickr under the Public Domain Mark 1.0 https://creativecommons. org/publicdomain/mark/1.0/). https://www.flickr.com/photos/144006675@N05/27487033282. https://www.flickr. com/photos/owen_lucas_photography/7454467582. https://www.flickr.com/photos/owen_lucas_photography/ 7454436922. https://www.flickr.com/photos/owen_lucas_photography/8102080226. Accessed on August 2016.
The following figure supplement is available for figure 1:

**Figure supplement 1.** Onset latencies and durations of gaps, voluntary and spontaneous blinks.

Materials and methods). Its end result is estimates of the contribution of each experimental event to the observed HFB timecourse over time, after the contributions of other experimental events were accounted for. See *Figure 3—figure supplement 1* for a demonstration of the advantage of this approach over standard event-related averaging.

## Observing gap and blink responses across the visual hierarchy

*Figure 4* presents examples of the gap-related, voluntary blink- and spontaneous blink-related responses in two pairs of cortical sites, each sampled within an individual patient. In both patient P46 (*Figure 4a*) and patient P39 (*Figure 4b*) there were concurrent recordings of ventral early (V1 and V2, respectively, defined by a surface-based probabilistic atlas, *Wang et al., 2014*) and ventral high-level (Ventral Occipital and Fusiform gyrus) visual sites. In both cases, activity in early visual sites was strongly modulated by blinks, undergoing a dip in activation levels following eye closure and then overshooting beyond expected activation levels as the eyes re-opened and the visual image reappeared. Gaps induced a qualitatively comparable effect – a negative dip in response to the onset of the display of the blank screen followed by a positive overshoot triggered by the reappearance of the stimulus. By contrast, the two higher-level ventral electrodes showed considerable reappearance-related responses following the gaps, but largely, no such responses to blinks. In P39, the high-level electrode was face-selective; by estimating blink and gap-related responses separately for face and non-face trials, we found that the reappearance-related overshoot induced by gaps (and to a lesser extent by voluntary blinks) in this electrode was face-selective as well.

Whereas most of our high-level visual electrodes in this study covered ventral regions, a similar yet distinct mode of transformation between early and late responses to blinks and gaps was evident in two cases with high-level dorsal coverage (*Figure 4—figure supplement 1*). Patient P57

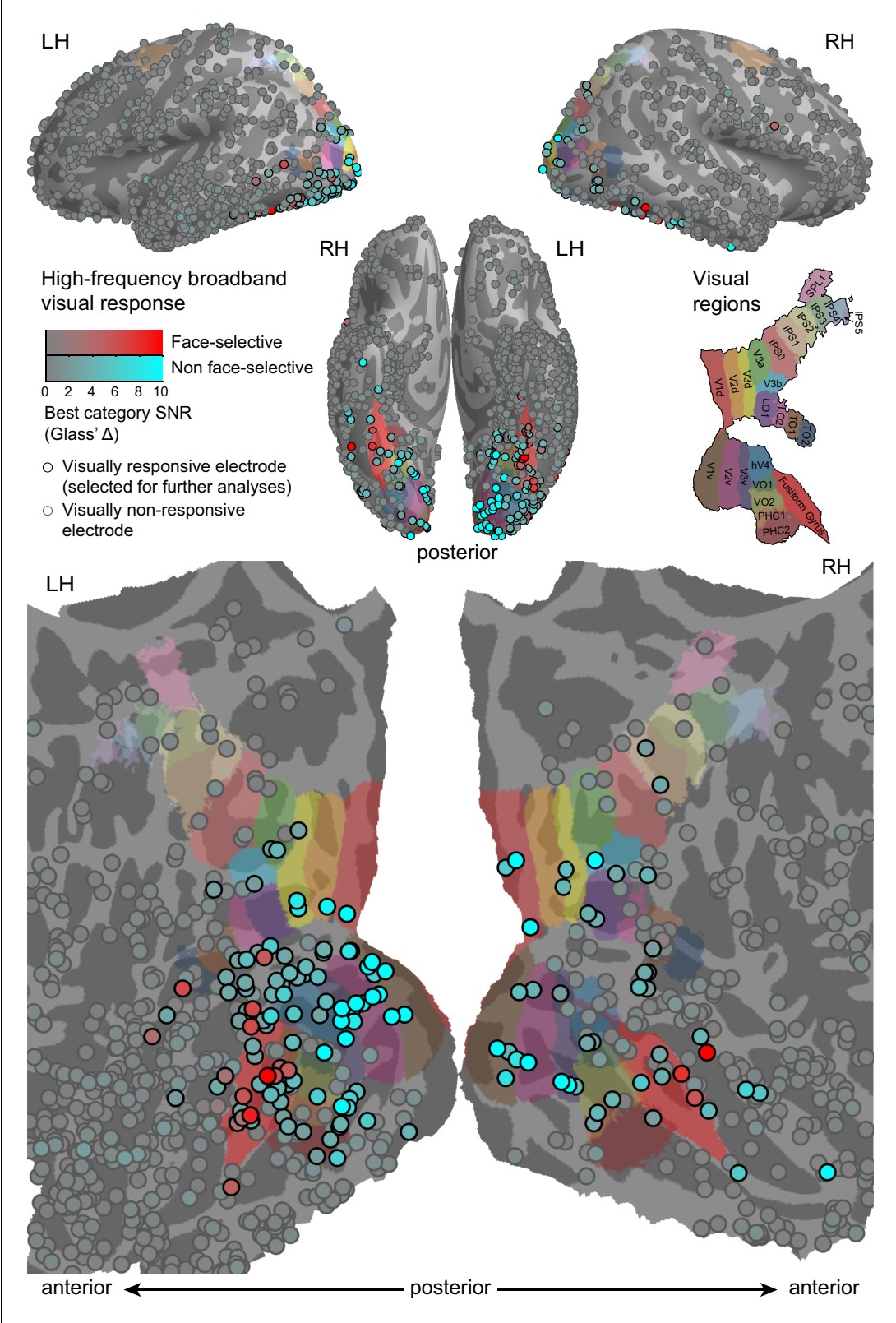

**Figure 2.** High-frequency broadband (HFB, 70–150 Hz) visual responses to object images. HFB responses from all participants, sampled from 50 to 350 ms following the transition from one object image to another, compared with the HFB activity sampled during inter-block blanks are presented. Each circle marks the location of one electrode on a common cortical template. The response strength for each electrode's optimal (maximally responding) category is measured in standard deviations of the baseline (Glass' Δ) and is color-coded as the circles' face color saturation. Electrodes showing

*Figure 2 continued on next page*

*Figure 2 continued*

significant face-selectivity are presented in a red hue and the others are presented in a cyan hue. Electrodes that passed the inclusion criteria for the subsequent analyses (corrected significance < 0.05 and an effect size of at least two standard deviations) are encircled in black. The colored labels on the cortical surface were derived from a surface-based atlas of retinotopic areas (*Wang et al., 2014*) and from Destrieux Atlas (*Destrieux et al., 2010*) as implemented in FreeSurfer 5.3 (Fusiform gyrus, in red).

The following source data is available for figure 2:

**Source data 1.** Individual electrode data for *Figure 2*.

(*Figure 4—figure supplement 1a,c*) had concurrent recordings in V1 and in a site in a high-level dorsal stream region, over the middle temporal gyrus, slightly less than 1 cm anterior to the border of anatomically defined MST/TO2. The dorsal high-level electrode showed a positive HFB response triggered by the image disappearance (*Figure 4—figure supplement 1a*, compare with **1c** for image-reappearance-lock) due to gaps and no response whatsoever for blinks. By contrast, both gaps and spontaneous blinks were associated with a considerable dip in V1 activation (that particular patient was not instructed to voluntarily blink). In patient P20 (*Figure 4—figure supplement 1b,d*), who had adjacent electrodes in anatomically defined dorsal V3 and V3a, a sharp short-range transformation was observed. The V3 site showed a response pattern compatible with the early sites in the previous three patients, whereas the V3a site showed strong positive responses both to stimulus disappearance (*Figure 4—figure supplement 1b*) and to stimulus reappearance (*Figure 4—figure*

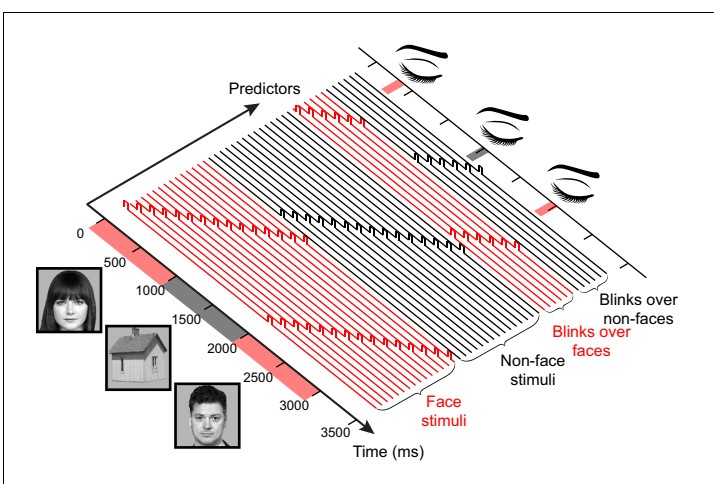

**Figure 3.** A schematic illustration of the general linear model (GLM) design matrix used in the deconvolution of the neural responses. The observed time series in each electrode is modeled as a linear sum of overlapping responses triggered by the displayed stimuli and by the different interrupting events – gaps, voluntary blinks and spontaneous blinks (for simplicity, only a single set of blinks predictors appears in the illustration). Each response is composed of a sequence of non-overlapping unit pulses (4 ms-wide pulses were used for the actual 250 Hz HFB timecourse, here a less detailed, 10 Hz model is presented). Note that in this example both the stimuli and the interruptions are modeled separately for face and non-face trials. See *Figure 3—figure supplement 1* for a demonstration of the advantage of this approach over standard-event related averaging.

Face photographs: Owen Lucas. Available on Flickr under the Public Domain Mark 1.0 https://creativecommons. org/publicdomain/mark/1.0/). https://www.flickr.com/photos/144006675@N05/27487033282. https://www.flickr. com/photos/owen_lucas_photography/7454467582. Accessed on August 2016.

The following figure supplement is available for figure 3:

**Figure supplement 1.** Bias in the estimation of gap-related responses due to unaccounted overlap of stimulus and gap responses and its correction by the deconvolution approach.

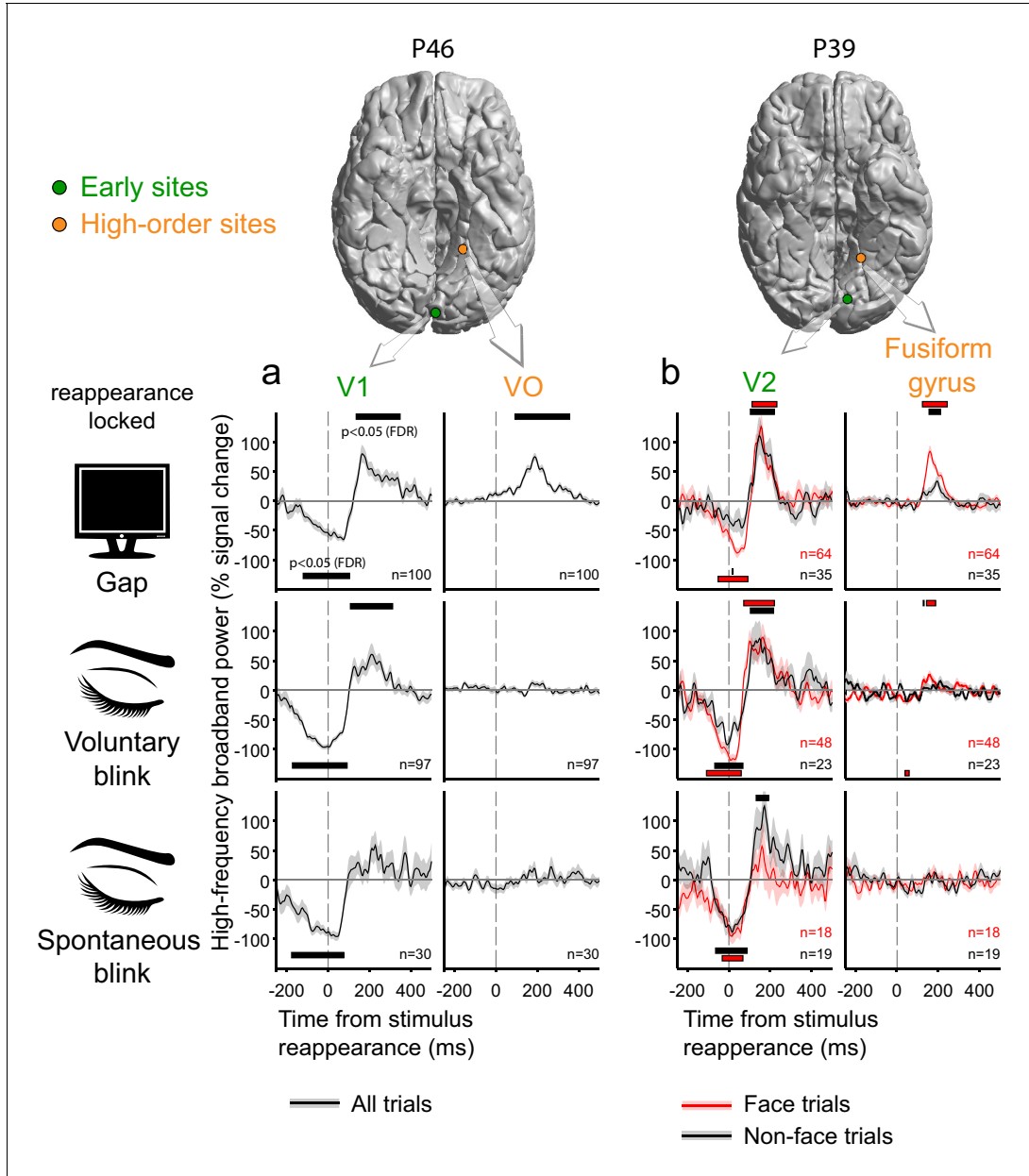

**Figure 4.** Deconvolved high-frequency broadband responses to gaps, voluntary and spontaneous blinks, simultaneously sampled at the early visual cortex and at the ventral high-level cortex. Two pairs of electrodes from two patients are presented. All responses here are locked to the stimulus reappearance. Error bounds show the standard error of the regression coefficients. Horizontal bars mark response timepoints significantly different from zero (p<0.05, FDR-corrected within-participant, see Materials and methods). n is the number of event occurrences. Note the activation-dip followed by a reappearance-related overshoot for gaps, voluntary blinks and spontaneous blinks in early visual sites in both participants. By contrast, the two ventral high-level sites showed almost no disappearance-related dip for all three interruptions and a reappearance-related overshoot response only following gaps. (a) In participant P46, events from all trials (face and non-face alike) are depicted in black. (b) In patient P39, the high-level electrode showed greater responses to faces. By estimating the contribution of gaps, voluntary and spontaneous blinks separately during face and non-face trials (red and black traces, correspondingly), it is evident that the high-level reappearance-overshoot effect is dependent on the category of the reappearing stimulus.

The following figure supplement is available for figure 4:

**Figure supplement 1.** Deconvolved high-frequency broadband responses to gaps, voluntary and spontaneous blinks, simultaneously sampled at the early visual cortex and at the dorsal high-level cortex.

*supplement 1d*) when these were caused by gaps but not when they were caused by voluntary or spontaneous blinks.

Moving from these four example electrode pairs to the entire sample of fourteen patients, we parceled the visually responsive electrodes into seven regions of interest (ROIs), ensuring that each ROI contained electrodes from at least six different patients. The resulting ROIs included five retinotopic regions, V1, V2, V3, V4 and VO, defined according to the surface-based probabilistic atlas (*Wang et al., 2014*), face-selective electrodes (defined functionally, see above) and high-level non-face selective electrodes defined as being situated outside and away of any of the retinotopic areas specified by the surface-based atlas (*Figure 5—figure supplement 1a*, see Materials and methods for electrode localization and parcellation details).

*Figure 5a* depicts the grand-averages of the deconvolved stimulus reappearance-related responses for gaps, voluntary blinks and spontaneous blinks within each ROI, separated into face and non-face trials (see *Figure 5—figure supplement 1b* for disappearance-locked grand-averages). Responses followed a consistent hierarchical progression: in V1 and V2, gaps, voluntary blinks and spontaneous blinks all produced qualitatively similar (but not identical) responses – which consisted of a dip in activation in response to the image disappearance, followed by an overshoot beyond baseline levels triggered by the image reappearance. Progressing along the visual hierarchy, the response to gaps and blinks diverged: the response to the image reappearance was sustained only when triggered by the termination of an external gap, and not by the termination of a blink. The negative activation dip reduced its amplitude along the hierarchy, even more so for gaps than to blinks. This is in stark contrast with the expected filling-in during blinks but not gaps. As an alternative analysis, we derived traditional event-related averages of the HFB signal of uninterrupted stimuli and subtracted them from each trial (instead of the deconvolution procedure) before computing gap and blink-related event-related-averages. This procedure yielded very similar results (*Figure 5—figure supplement 1c*), which rules out the possibility that the observed results were somehow an artifact of the deconvolution analysis we adopted.

*Figure 5b* depicts the response to the object images themselves, after accounting for the effects of gaps and blinks. Note the considerable decline within several hundred milliseconds in response amplitude. In order to examine the possible effect of this decline on the HFB-dips, gap-related HFB responses were grouped by the gaps' latency relative to trial onset (*Figure 5—figure supplement 2*). The result of this analysis revealed similar dip magnitudes for gaps happening earlier and later in the trials in high-order regions.

In order to provide a more direct visualization of the data, we generated videos depicting the reappearance-locked responses to gaps (*Video 1*) and spontaneous blinks (*Video 2*) timepoint by timepoint, across all of the 143 electrodes. Following the gaps' termination, a wave of activation that began in V1 and spread up to the anterior edge of the visual cortex, can be observed. By contrast, the blinks' termination triggered a much more localized positive activation that remained confined to V1–V3.

## Quantifying single electrode negative dip and positive overshoot components

In order to statistically test the reproducibility and generalizability of the observed response patterns, we quantified the two most recurring components of the gap and blink-related responses, the negative activation dip following image disappearance (interruption onset) and the positive overshoot following the image reappearance (interruption offset), on an individual electrode basis. Specifically, for each electrode and interrupting event (gap/spontaneous blink/voluntary blink), we searched the reappearance-related response for clusters of contiguous above-baseline timepoints. The cluster of largest activation integral that followed the reappearance was marked as the electrode's reappearance-related overshoot component. Similarly, the contiguous below-baseline cluster in the disappearance-locked response with the largest (negative) activation integral was marked as the electrode's negative-dip component. This procedure directly addressed the response-latency variability across electrodes, reflected in the travelling wave-like nature of the responses across the visual hierarchy (see *Videos 1* and *2*).

The effects within each estimated component were statistically tested by two complementary approaches: the first was multiple-comparisons-corrected permutation testing in each individual electrode, as commonly done in electrocorticographic (ECoG) studies. The second was an ROI based

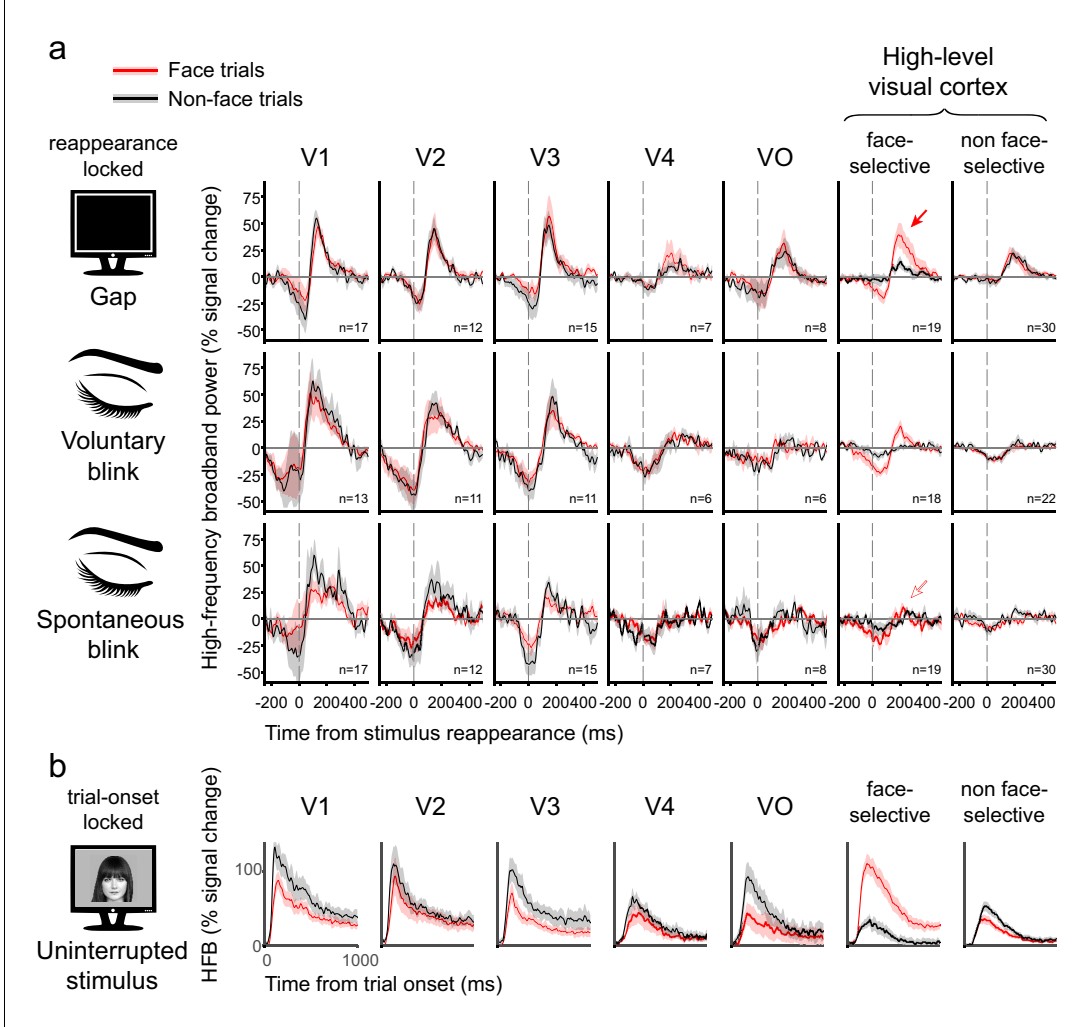

**Figure 5.** Grand averages of deconvolved high-frequency broadband responses. (**a**) Responses to gaps, voluntary blinks and spontaneous blinks along the visual hierarchy. All of the traces here are locked to the image reappearance, marked as t = 0. See *Figure 5—figure supplement 1b* for the analogous stimulus-disappearance locked traces. After within-electrode estimation, the traces were averaged first within individuals and then across individuals, such that each grand-average is derived from independent individual traces. n is the total number of averaged electrodes for each trace. Error bounds show the standard error of the mean across individuals. Note the gradual appearance of differential responses to gaps compared with voluntary and spontaneous blink as the visual signal traveled forward. (**b**) Responses to object images (face and non-faces). Note that each stimulus was presented for one whole second.

Face photograph: Owen Lucas. Available on Flickr under the Public Domain Mark 1.0 https://creativecommons.org/publicdomain/mark/1.0/). https://www.flickr.com/photos/144006675@N05/27487033282. Accessed on August 2016.

The following figure supplements are available for figure 5:

**Figure supplement 1.** ROI parcellation and additional grand-averages.

**Figure supplement 2.** Effect of gap latency.

mixed-effects group analysis directly assessing the generalizability across electrodes and patients' ROIs. It should be emphasized that in both tests, the potential effect of selection bias (commonly referred to as 'circular analysis') on these measurements was eliminated. In the permutation tests, both the real (unshuffled) and permuted data were subject to the same level of selection bias. The ROI-based mixed-effects group analysis was performed on unbiased response estimates derived by a split-half approach (see Materials and methods).

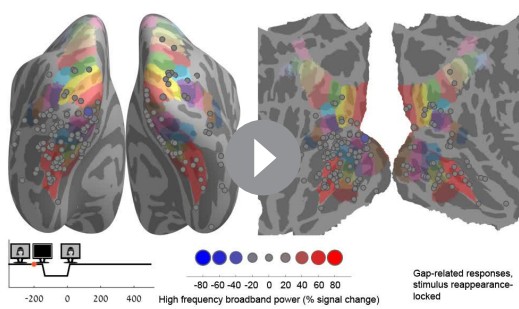

**Video 1.** Response to gaps (stimulus-reappearance lock).
Face photograph: Owen Lucas. Available on Flickr under the Public Domain Mark 1.0 https://creativecommons.org/publicdomain/mark/1.0/). https://www.flickr.com/photos/144006675@N05/27487033282. Accessed on August 2016.

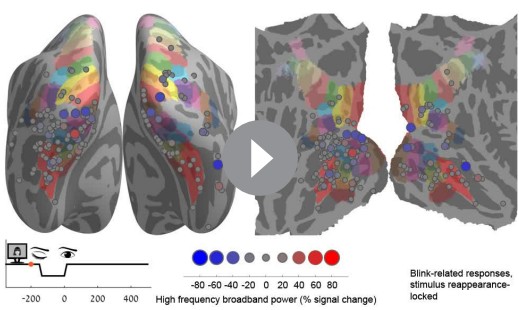

**Video 2.** Response to spontaneous blinks (stimulus-reappearance lock).
Face photograph: Owen Lucas. Available on Flickr under the Public Domain Mark 1.0 https://creativecommons.org/publicdomain/mark/1.0/). https://www.flickr.com/photos/144006675@N05/27487033282. Accessed on August 2016.

## Electrode-level statistical analysis

In accordance with the pattern seen in the grand averages, significant electrode-level differences between the reappearance-related overshoot responses for gaps and spontaneous blinks emerged in higher-level visual cortex (*Figure 6*). 51 sites (out of 143) showed a significant advantage for gaps over spontaneous blinks (two-tailed permutation test, $p_{FDR} < 0.05$). The effect was not uniformly distributed across the ROIs: V1 showed no significant electrodes out of 17 electrodes, V2 – 0/12, V3 – 5/15, V4 – 2/7, VO – 4/8, face-selective – 10/19 and high-level face non-selective ROI – 18/30. Randomization test of independence confirmed the apparent inhomogeneity of effect occurrence ($\chi^2(6) = 17.219$, p=0.00001). Comparing voluntary blinks with gaps (*Figure 6—figure supplement 1*) revealed a similar, but weaker pattern: 35 of 108 electrodes recorded in the 10 patients who performed voluntary blinks showed significant advantage of gaps over voluntary blinks (two-tailed permutation test, $p_{FDR} < 0.05$), and the effect was more prevalent in higher-level ROIs: V1 – 1/13, V2 – 1/11, V3 – 2/11, V4 – 1/6, VO – 4/6, face selective – 10/18, face non-selective – 13/22 ($\chi^2(6) = 13.137$, p=0.001). No electrodes showed significantly greater reappearance-related overshoot for voluntary blinks than for gaps.

For the activation dip component, there was evidence of dips significantly greater for spontaneous blinks than for gaps in 26 of 143 electrodes (two-tailed permutation test, $p_{FDR} < 0.05$), found mostly in early visual cortex – V1 – 6/17, V2 – 7/12, V3 – 2/15, V4 – 3/7, VO – 1/8, face-selective 0/19 and high-level non-face selective 4/30 ($\chi^2(6) = 16.498$, p=0.001). No electrode showed significantly greater dips for gaps than for spontaneous blinks, again a result inconsistent with the selective filling-in hypothesis. Voluntary blinks showed higher occurrence of significantly greater dip for blinks than for gaps (51 of 108 electrodes), with no evidence for non-uniform distribution across ROIs: V1 – 9/13, V2 – 8/11, V3 – 7/11, V4 – 3/6, VO – 4/6, face-selective – 7/18, high-level face non-selective – 11/22 (Randomization test of independence n.s., $\chi^2(6) = 2.299$, p=0.53). No electrode showed evidence of greater dips for gaps than voluntary blinks.

## Mixed-effects ROI based statistical analysis

In order to test the generalization of these results across electrodes and patients, we fitted the reappearance-related overshoot component (and alternatively, the disappearance-related activation dip component) with a three-way mixed-effects model with the fixed factors of trial-category (face/non-face), interruption type (gap/voluntary blink/spontaneous blink) and ROI and the random factors of electrode and patient. The results of this analysis can be interpreted similarly to the more commonly used repeated-measures ANOVA, however, since the mixed-effects approach naturally allows for missing values, it is well suited for data sampled with varying anatomical coverage. The estimated responses are presented in *Figure 7*. We first describe the results for the reappearance-related overshoot (*Figure 7a–b*). The two ANOVA tests of primary interest are (a) the interaction of ROI with

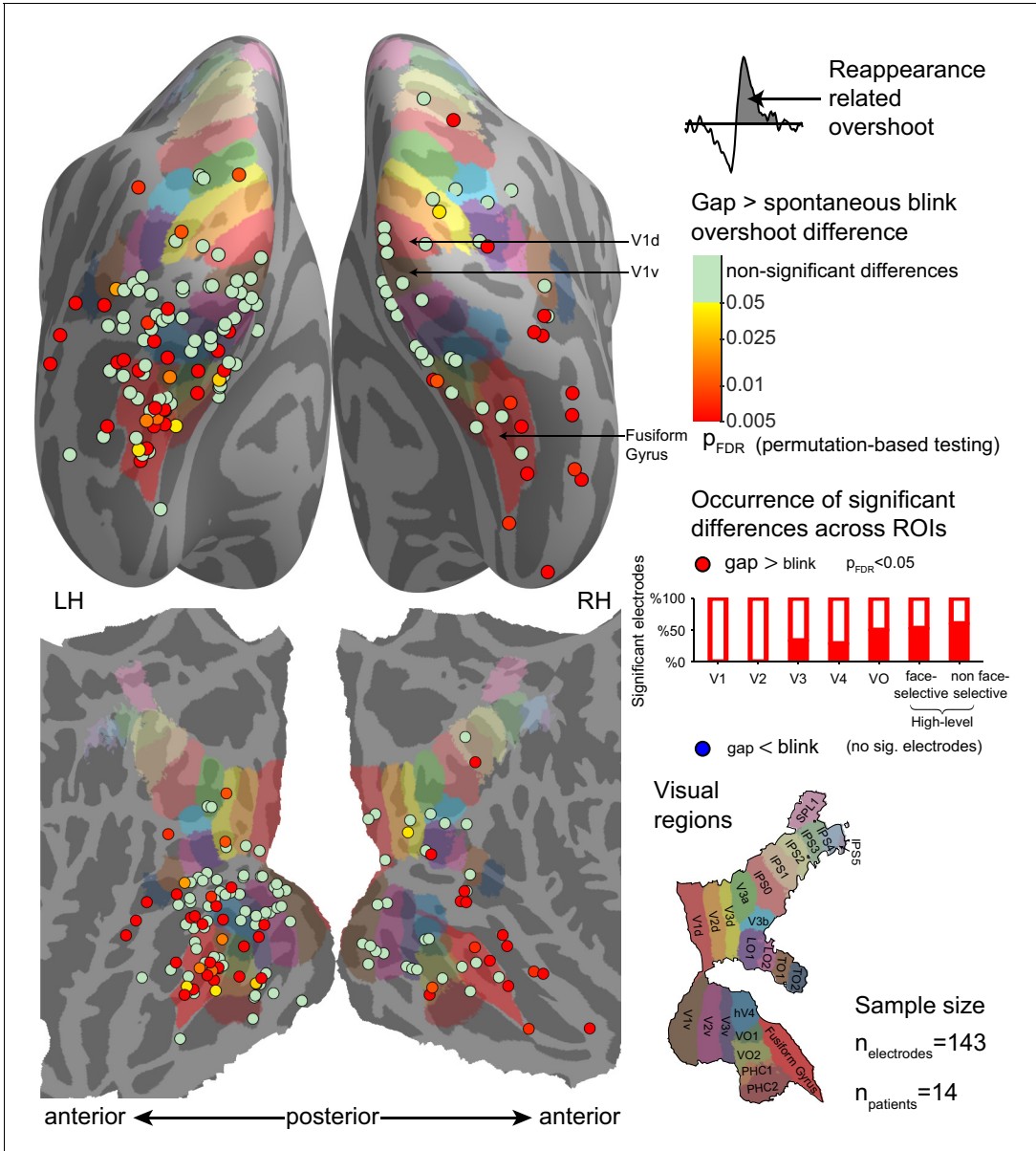

**Figure 6.** Electrode-level permutation testing of the reappearance-related response overshoot for gaps compared with the same measure for spontaneous blinks. See *Figure 6—figure supplement 1* for a comparison of gaps with the voluntary blinks. Each circle stands for one of 143 visually responsive electrodes polled across 14 participants, colored according to the FDR-adjusted permutation test p-value (logarithmic color scale). The partially filled bars on the right show the percentage of electrodes showing a significant gap overshoot > blink overshoot effect within each region of interest. None of the visually responsive electrodes exhibited a significant inverse effect (blink overshoot>gap overshoot).

The following source data and figure supplements are available for figure 6:

**Source data 1.** Individual electrode data for *Figure 6*.

**Figure supplement 1.** Electrode-level permutation testing of the reappearance-related response overshoot for gaps compared with the same measure for voluntary blinks.

**Figure supplement 1—source data 1.** Individual electrode data for *Figure 6—figure supplement 1*.

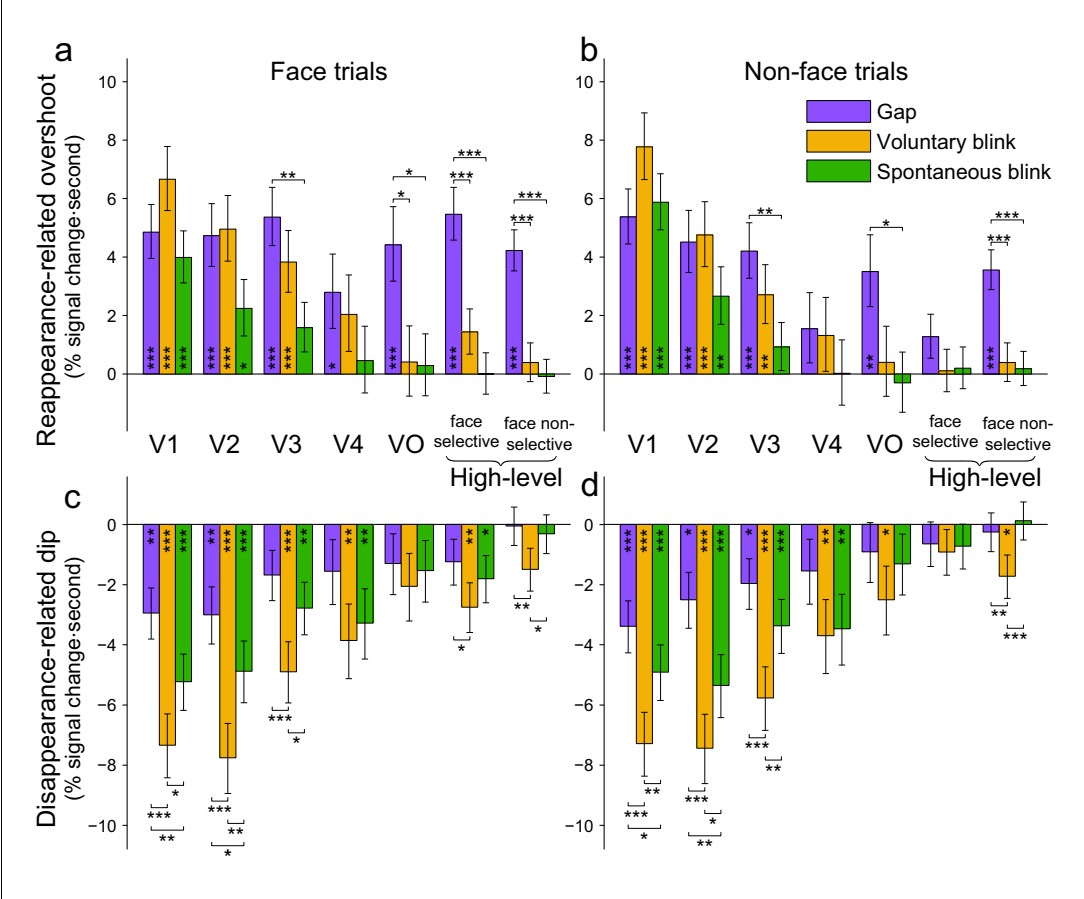

**Figure 7.** Mixed-effects response estimates. The bars depict estimated average magnitudes (mixed-effects least squares means and their SEs) of reappearance-related overshoots (upper panels) and disappearance-related activation-dips (lower panels) for gaps, voluntary blinks and spontaneous blinks, occurring during face trials (left panels) and during non-face trials (right panels). Asterisks mark significant within-ROI simple effects ($p_{FDR} < 0.05$ – *, $p_{FDR} < 0.01$ – **, $p_{FDR} < 0.001$ – ***). (a) Average reappearance-related overshoot magnitudes (face trials). (b) The same measurement for events occurring during non-face trials. (c) Disappearance-related activation-dips magnitudes (deeper activation-dips are more negative) during face-trials. (d) The same measurement for events occurring during non-face trials.

The following source data and figure supplements are available for figure 7:

**Source data 1.** Mixed-effect model outputs used to create *Figure 7*.
**Figure supplement 1.** Mixed-effects group analysis of gap and blink events matched for onset latency.
**Figure supplement 2.** Mixed-effects group analysis of gap and blink events matched for duration.
**Figure supplement 3.** Controls for gap low-level properties.

interruption, which was significant ($F(12,471.29) = 4.38$, $p<0.001$), indicating that gaps, voluntary blinks and spontaneous blinks indeed affected different ROIs differently and (b) the three-way interaction between ROI, interruption and category, which was not significant ($F(12,463.31) = 0.59$, $p=0.85$), potentially because category appeared to modulate interruption only in a single ROI (face-selective electrodes). The significant ROI and interruption interaction was further investigated by testing for simple effects between spontaneous blinks, voluntary blinks and gaps within each ROI. In agreement with our previous analysis, in face-trials, gaps showed significantly larger reappearance-related overshoot responses compared with voluntary blinks both in VO, the face-selective and high-level face non-selective ROIs ($p_{FDR} < 0.05$). Compared with spontaneous blinks, gaps had

significantly larger reappearance-related overshoot response also in V3 ($p_{FDR} < 0.05$). In non-face trials (*Figure 7b*), gaps had a significantly larger overshoot response compared with voluntary blinks in the high-level face non-selective ROI. Compared with spontaneous blinks, gaps had significantly larger reappearance-related overshoot (in non-face trials) also in V3 and VO.

Fitting the disappearance-related activation-dip component with the same mixed-effects model (*Figure 7c–d*) also found a significant interaction of ROI and interruption (F(12,466.48) = 4.26, p<0.001). As in the reappearance-related overshoot data, the three-way interaction of interruption, category and ROI was not significant (F(12,463.10) = 0.45, p=0.94). Inspecting the effects between the three kinds of interruptions within each ROI found that the activation dip triggered by gaps was smaller (more shallow) compared to that triggered by voluntary blinks in most ROIs (see *Figure 7c–d* for detailed simple effects results, $p_{FDR} < 0.05$) and it was also smaller than the dip triggered by spontaneous blinks in V1 and V2 (both face and non-face trials, $p_{FDR} < 0.05$). Importantly, no evidence was observed for greater (deeper) activation dip for gaps compared with either spontaneous or voluntary blinks in any ROI, and except for the high-level face non-selective ROI during non-face trials, the non-significant trend in all ROIs was inverse (gaps had smaller dips than both kinds of blinks), once again, a result that is inconsistent with selective filling-in of blinks but not of gaps. Repeating the mixed-effects analysis after normalizing each ROI by its total response to the object-images (e.g. normalizing by the integrals of the traces depicted in *Figure 5b*) yielded very similar results. Note that both the electrode-level permutation testing and group mixed-effects analysis demonstrated a compatible pattern of effects.

## Controls for gap duration and latency

Since the limitations involved with the clinical setting did not allow us to present the gaps in an exact physical replay of the individual patients' blink latencies and durations, differences in these properties could have confounded our results. We tested that possibility by post-hoc selecting subsets of gaps and blinks such that their latency or duration histograms will tightly match (*Figure 7—figure supplements 1,2*, correspondingly). The matching procedure was done independently for voluntary and spontaneous blinks, using face-trials. Whereas this post-hoc selection led to considerably increased sampling error estimates (due to the usage of only a fraction of the original trials), the observed pattern of results largely reproduced the results reported above and was qualitatively indistinguishable from that produced by the entire dataset, both for matching the events latencies (*Figure 7—figure supplement 1c–f*) and for matching the events durations (*Figure 7—figure supplement 2c–f*). Specifically, in all matched analyses, we still observed significant advantages of the reappearance-related overshoot for gaps compared with spontaneous and voluntary blinks in the higher-level ROIs (face-selective and non-face selective high-level visual electrodes, $p_{FDR} < 0.05$). The finding of greater (deeper) activation dip for blinks compared with gaps was even more pronounced than in the original (unmatched) analysis, with no ROI showing even a non-significant trend in the inverse direction.

To further control for potential low-level differences between the spontaneous blinks and gaps we included two additional experimental controls. First, in nine patients, half of the gaps were implemented with a gray instead of a black blank. This manipulation has the effect of decreasing both the amplitude of the image transients caused by the gap and their spatial extent; since the same gray-level was used as the screen background, only the central part of the screen was perturbed instead of its entirety. If transients produced by gaps persisted in higher-level visual cortex due to greater low-level impact, this manipulation should reduce this effect. However, using gray instead of black gaps produced qualitatively similar results (*Figure 7—figure supplement 3a–b*), keeping the pattern of high-level advantage of gaps over blinks intact.

A second control, conducted in seven patients, consisted of displaying half of the gaps gradually, fading through three frames of 25%, 50% and 75% intermediate mixing levels (manipulated orthogonally of the gray/black gap manipulation), each fade-in or fade-out lasting a total of 66 ms. Similarly to the gray-gaps manipulation, this lower-level manipulation of the gaps did not alter the results qualitatively (*Figure 7—figure supplement 3c–d*).

To control for potential blink artifacts, the ECoG analysis pipeline (HFB computation followed by deconvolution) was applied also to the EOG channel. This analysis found a positive blink-related artifact that peaked roughly at the same time as our eye-tracker derived blink onsets (mean±SD 1.43 ms ± 23.8) and about 130 ms (−134 ms ± 30.2) before the eye tracker derived blink offsets.

There were no negative HFB responses in the EOG channels, excluding an artifactual source for the ECoG blink-related HFB-dips.

## Discussion

### The main findings

By inspecting the high-frequency broadband responses to visual images interrupted by spontaneous blinks, voluntary blinks and blank frames (gaps), we observed three main results: (a) early visual cortex (V1 and V2) responded in a similar fashion to blinks and gaps: in both cases, HFB activity dipped following the stimulus disappearance and then increased beyond the uninterrupted activation levels, triggered by the stimulus' reappearance. (b) The activation overshoot triggered by blinks' termination gradually decreased along the cortical hierarchy, whereas the overshoot produced by gaps' termination was sustained, resulting in an increasing difference between the responses to gaps' and blinks' termination in higher-level visual cortex. (c) We found no supporting evidence for the filling-in hypothesis, which predicts a sustained activation during blinks but not during gaps; rather, our results showed an inverse trend in most of the ROIs.

### Potential confounds

Due to technical limitations imposed by the clinical setting, the gaps employed in the current study were only an approximate simulation of the retinal impact of eye blinks. Some of the potential low-level differences between these two conditions concerned the events' durations, their latency in relation with the trials' onsets, their luminance levels, their visual field extent and the relative abruptness of their onsets and offsets. Our series of controls argue against the possibility that these low-level differences could account for the observed effects – neither the direct matching of gaps and blinks by duration or latency nor the deliberate manipulation of the other stimulus aspects qualitatively altered the observed pattern of results (see *Figure 7—figure supplements 1–3*).

Nevertheless, even after considering these controls, the gaps and the blinks are still distinct physical events with retinal impacts that cannot be assumed to be precisely equivalent. Can these potential retinal differences account for the differential response to gaps and blinks we observed in high-level visual cortex? We believe that the observed similarity between gap and blink-related responses in early visual cortex empirically argues against this possibility. Put simply, the responses in early visual cortex can be viewed as an 'internal control' of the low-level effects of these two types of events- and these early responses were quite similar for gaps and blinks (see *Figure 5a*, left column). This interpretation is supported by a large body of data demonstrating the preferential sensitivity in early visual cortex to low-level aspects of the stimulus compared to higher order representations (for a review, see *Grill-Spector and Malach, 2004*).

It could be argued that the lack of differential HFB-dip for gaps compared with blinks in high-order regions may be due to the substantial decline in the sustained responses to the object images in these regions (*Figure 5b*). Such a decline could have resulted in a 'floor-effect' for those gaps and blinks that occurred later in the trial when the signal was already low. However, early and late-onset gaps showed similar depression of activity following image disappearance (*Figure 5—figure supplement 2*).

Oculo-motor-related effects (resulting from eye-muscle activity or eye and eyelid movements) can represent another potential source of artifactual differences in ECoG signals between blinks and gaps. In scalp EEG recordings, such effects are indeed a major concern. However, previous research has demonstrated that in intracranial recordings these artifacts are limited to regions more anterior than the visual responsive sites that were used in our study (*Ball et al., 2009*). Moreover, even without relying on the electrodes' anatomical locations, the timing of our reported effects, following the blinks by tens to hundreds of milliseconds, unequivocally indicates their neural source (see Results).

Lastly, the different number of gap and spontaneous blink events might have posed a statistical issue. Importantly, blink and gap-related responses were tested against each other, a comparison that is not biased by SNR differences. Furthermore, the pattern of results was unaltered even when the number of events was equated (*Figure 7—figure supplements1,2*).

## Theoretical implications

### Early visual cortex responds similarly to perceptually distinct events

The striking similarity in the response profile of early visual electrodes between the spontaneous blinks and the gaps, despite their marked perceptual difference, reinforces single-unit results from non-human primates comparing reflex-blinks with external darkenings (*Gawne and Martin, 2000, 2002*) and strongly supports the notion that early visual responses are driven mainly by the optical stimuli impinging on the retina and are minimally related to their perceptual status, when the low-level inputs are similar. This appears to be compatible with a range of previous observations unrelated to eye blinks (see *Rees et al., 2002*; *Grill-Spector and Malach, 2004*), including individuals with early visual cortex lesions that nevertheless report vivid visual experiences during dreaming (*Solms, 1997*) or even while awake (*Ashwin and Tsaloumas, 2007*). The present results extend this conclusion to the realm of supra-threshold and naturalistic sensory stimulation.

### High-level visual cortex shows differential responses to perceived vs. unperceived perturbations

In higher-level visual cortex (mostly ventral sites, in our sample), the perceived reappearance of the object image following the termination of gaps triggered a HFB overshoot that was absent for the unperceived image-reappearance that followed spontaneous blinks. This result reinforces the ecological validity of the view that bursts of high-frequency broadband activity in ventral higher-level cortex are correlated with the appearance of perceived visual object images, previously demonstrated using backward-masked near-threshold stimuli (*Fisch et al., 2009*; *Moutard et al., 2015*).

Comparing gaps with voluntary blinks revealed a similar yet less pronounced effect, i.e., a more moderate distinction from gaps relative to spontaneous blinks. This may reflect the intermediate perceptual status of voluntary blinks, visible but systematically underperceived (*Riggs et al., 1981*). An alternative possibility may attribute such differences to the greater duration of the voluntary blinks evident in many of the patients (*Figure 1—figure supplement 1*). However, the replication of the results by the matched-gap-durations control (*Figure 7—figure supplement 2*) suggests that such an effect of duration on higher-level visual regions may be minimal.

Given the evidence from the early visual cortex of a similar retinal impact of blinks and gaps, the most likely (but not exclusive, see below) interpretation of the reduction in blink-related transients in higher-level visual cortex is extra-retinal suppression of the cortical neural response to the retinal impact of blinks. Whereas studies neutralizing blinks' retinal impact have demonstrated a certain degree of extra-retinal blink-related effects already in V1 (*Buisseret and Maffei, 1983*; *Bristow et al., 2005b*), our results suggest that the effective locus of the suppression of neural responses to blinks is likely further up the visual hierarchy. It may be that the differences between the responses to blinks and external darkenings that were observed in a small minority of early visual cortex single units in the macaque (*Gawne and Martin, 2000, 2002*) may act as precursors to the differential reappearance-related response at the population level we observed in human higher-level visual cortex.

How does the visual cortex receive the information that discriminates between gaps and blinks? Our observation of reduced blink-related responses reflects the end-product of a putative blink-suppressing signal. An obvious candidate of such a signal is that of 'corollary discharge' or 'efference copy' – a copy of the motor command to execute the blink that is sent in parallel to the visual cortex, via a motor-sensory pathway (see *Wurtz, 2013*). This pathway might be shared with saccades: this possibility is supported by psychophysical similarities of blink-suppression and saccade-suppression (*Bidder and Tomlinson, 1997*) and by the documented occurrence of a brief 1–2° rotation of the eye during the downward phase of the blink, which might provide a sufficient efferent signal to enable blink suppression (*Riggs et al., 1987*). Alternatively, given the dramatic retinal impact of eyelid closure, it seems plausible that at least one source of this motor-sensory pathway should be blink-specific, hypothetically originating at the facial motor nucleolus, where the lower motor neurons controlling eyelid closure are located.

The potential source of the suppressing signal was not observed in the current study: neural activity that reliably preceded the execution of blinks was not detected in any of our recording sites. This might be related either to our incomplete coverage of oculomotor cortical regions or to the

potential sub-cortical locus of the (spontaneous) blink generator (*Kaminer et al., 2011*), which was obviously not covered by our recordings.

An alternative possibility to efference copy is that the discounting of responses to blinks is achieved by a more general-purpose prediction error minimizing mechanism (e.g. *Bastos et al., 2012*). Further research is needed to distinguish between these putative mechanisms.

An additional potential factor that may be involved in the observed reduction of blink-related transients is neural adaptation. Consider the following scenario, not tested by the current experiment, in which the presented stimulus is swapped during the blink. Would the new stimulus induce a reappearance-related overshoot as in gaps? A positive answer to this question would indicate that gaps (but not blinks) cause a release from adaptation, offering a rather different (yet extra-retinal) explanation of the high-level differential response to gaps and blinks.

Last, it is worth considering that the perceived continuity across blinks may rely on aspects of neural activity that were not probed by the current recordings. For example, while HFB's ignition-like dynamics are consistent with the timescale of iconic memory, ongoing maintenance of information

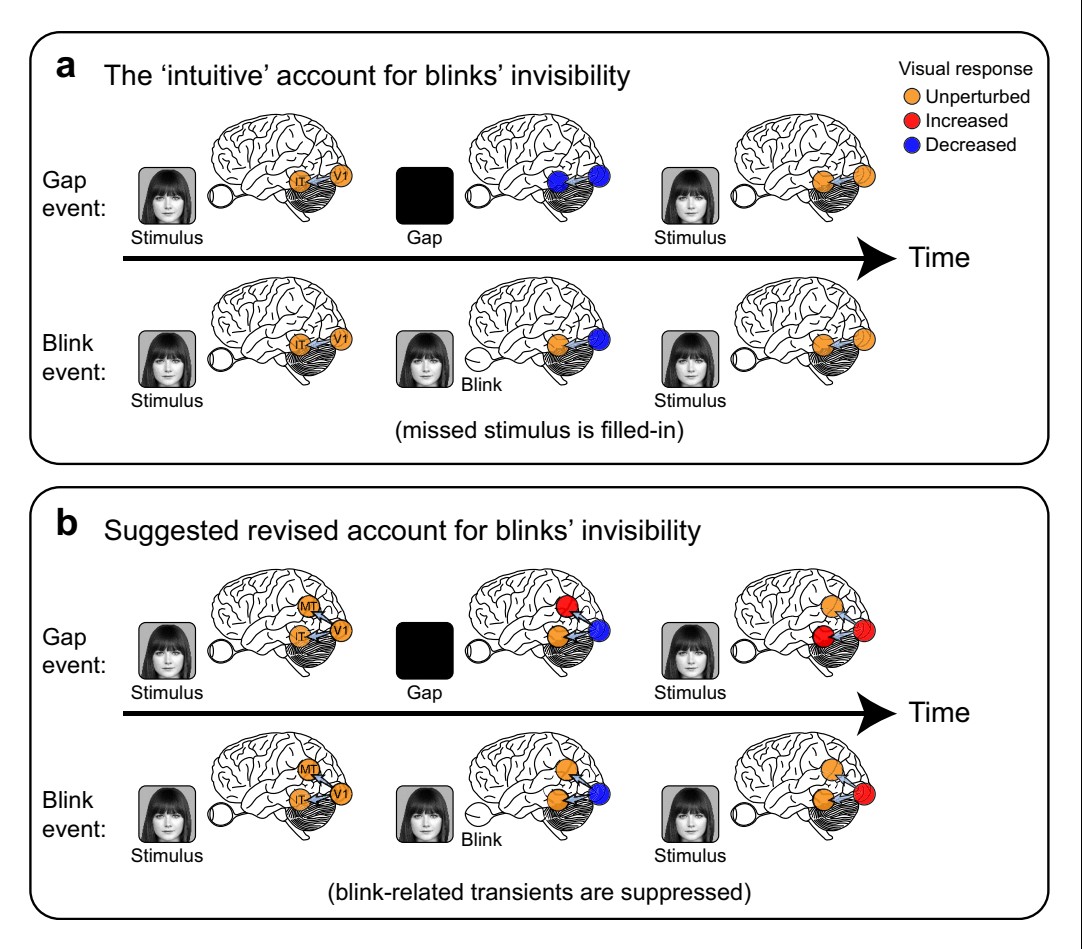

**Figure 8.** Two alternative accounts for the perceived continuity across blinks compared with perceived discontinuity following external blanking of the stimulus ('gaps'). (a) The 'intuitive' or commonsense account, by which the occluded face-stimulus is actively 'filled-in' by ventral high-level, category-selective visual regions during blinks but not during gaps, resulting in perceived continuity only across blinks. (b) A suggested revised account, consistent with the current findings: transient neuronal activations follow the perceived disappearance and reappearance of the face-stimulus. These transients (red filled circles) are evident in gaps but are extra-retinally suppressed in blinks, forming a distributed neural correlate of the perceptual distinction between these two events.
Face photograph: Owen Lucas. Available on Flickr under the Public Domain Mark 1.0 https://creativecommons.org/publicdomain/mark/1.0/). https://www.flickr.com/photos/144006675@N05/27487033282. Accessed on August 2016.

across larger timescales (as in short-term memory) is not correlated with HFB increases (e.g., *Noy et al., 2015b*).

### Distributed neural correlates of disappearance versus continuity

The sum of our findings suggests a refined view of the perceptual representation of brief stimulus disappearances. It is usually taken for granted that the perceived absence of a stimulus is a product of the absence of a corresponding stimulus-related neural activation. Such a commonsense model suggests that stimuli do not seem to disappear during spontaneous blinks because the stimulus-related activation in ventral high-level visual cortex is filled-in, whereas perceived disappearances leave 'neural gaps' in these representations (*Figure 8a*). However, our findings are inconsistent with this intuitive model. As our results show, brief disappearances, even when perceived, appear not to be faithfully represented as neural gaps in ventral high-level visual areas (*Figure 5a*). A plausible neural mechanism that may underlie such continuity of neural activity could be the self-reverberatory dynamics in ventral high-order regions that lead to a sluggish all-or none response (*Fisch et al., 2009*; *Moutard et al., 2015*).

Additional evidence for a dissociation between momentary activity reductions and perceptual disappearance in high-order visual areas can be found in the consistent and rapid decline we observed in the HFB activity in these regions during the one-second long display of the stimulus (*Figure 5b*), an effect with no apparent perceptual correlate.

Thus, it is interesting to note that the occurrence of signal decline without perceptual disappearance (*Figure 5b*) and perceptual disappearance (gap) without signal decline (*Figure 5a*) complement each other in suggesting that signal reduction on its own may simply fail to register perceptually. It is tempting to speculate based on these converging phenomena that perhaps the perceptually-relevant signals, at least in high-order visual areas, are phasic bursts rather continuous activations. This raises an intriguing issue – how is perceived stimulus disappearance actually represented?

One possibility is that the perception of brief stimulus disappearance may be reliably represented by a positive signal. It is conceivable that the onset (disappearance-related) and offset (reappearance-related) sensitive responses in higher-level dorsal stream sites that we noted (albeit infrequently due to coverage limitations. See *Figure 4—figure supplement 1*) play a critical role in this task: a burst of activity in such sites may underlie the percept of a flicker caused by a stimulus disappearance (gap) or external darkening. By contrast, during blinks, blink-suppression signals may actively block these detectors of discontinuity, resulting in the experience of continuously perceived stimuli (*Figure 8b*). Thus, we propose that the neuronal representation of perceived face-disappearance is found outside the ventral face-selective areas. The upshot of this is that one cannot attribute the content of a conscious percept to a single cortical element, necessitating a joint-representation across diverse content-selective regions.

# Materials and methods

## Participants

Intracranial recordings were obtained from fourteen individuals (four females, mean±SD age 34 ± 10.7), monitored for pre-surgical evaluation due to pharmacologically intractable epilepsy (see *Table 1* for individual demographic, clinical and experimental details). All patients gave fully informed consent, including consent to publish, according to NIH guidelines, as monitored by the institutional review board at the Feinstein Institute for Medical Research, in accordance with the Declaration of Helsinki. Data was obtained as part of protocol number 07–125. Patients had the opportunity to consent prior to electrode implantation and were informed that they may choose to decline or later withdraw from the study without affecting their clinical care. Consent includes agreement to participate with studies of cognitive and sensorimotor processes and publication of any deidentified data obtained. Risks include tedium and potential breach of medical information and are minimized by giving ample breaks and implementation of protocols to deidentify data close to the time of recording. Benefits to the subject include increased monitoring of the electrocorticogram and involvement of research methods to help localize electrodes with respect to preoperative MRI. Five additional patients performed the experiment but their recordings were excluded after initial data quality inspection. Reasons for exclusion were independent of the main analyses and consisted of:

(1) widespread interictal spikes, (2) no visual coverage, (3) severe visual field impairments (4) contamination of the ECoG signal by synchronization triggers and (5) a case in which the patient misunderstood the instructions to voluntary blink and executed prolonged eye closures (mean duration of 642 ms) instead.

## Experimental task

The subjects were seated in bed in front of an LCD monitor, approximately 70 cm from the screen. Blocks of grayscale still images of faces, houses, tools, abstract patterns or animals, embedded in a uniform gray background, were presented at a constant pace of 1 Hz, with no blanks between consecutive stimuli. The images subtended a visual angle of 15.8°. Stimuli were grouped in blocks of ten consecutive images of mixed categories, separated by between-block intervals of blank gray screen (three seconds long). A total of 32 blocks per subject were considered for this study (additional 12 blocks included sudden spatial displacements of the stimuli and were not included in the analysis). In 10 of the 14 patients (see *Table 1*), twelve of the blocks were preceded by an auditory cue to blink at a steady pace of one blink per second until a stop signal that followed the block's end. Blinks occurring during these blocks were categorized as 'voluntary' and blinks occurring outside these blocks were classified as 'spontaneous'. In the rest of the blocks, gaps in the visual stimulation were pseudo-randomly introduced in 60% of the trials at 350, 550 or 750 ms post stimulus transition. We applied a wide range of gap durations (100–200 ms in most patients, see *Figure 1—figure supplement 1*) in order to allow for post-hoc matching of blinks and gaps. The subjects' main task was to click the mouse button when an image of an animal appeared. These trials were later excluded from the analyses. Each patient participated in a single session that was divided by three short breaks in which the patients were required to rest with their eyes closed. Including the breaks, an entire session typically lasted about 15 min. As the experiment proceeded, additional controls for the gap event were introduced: In nine of the subjects (starting at P36), half of the gaps were gray (intensity level 183/255, where the object stimuli mean intensity level was 160/255) instead of black (0/255 intensity level) and in seven of these patients (starting at P46), half of the gaps were faded in and out through four frames (16.6 ms each) of 25%, 50% and 75% and 100% mixing levels between the image and the blank screen (manipulated orthogonally of the gray/black gap manipulation). The task was implemented using Presentation (Neurobehavioral Systems).

## Electrode implant and data acquisition

Recordings were conducted at Northshore University Hospital, Manhasset, NY, USA. The patients were implanted with subdural grids, strips, and/or depth electrode shafts (Ad-Tech Medical Instrument, Racine, Wisconsin). In the subdural grids and strips, each recording site was 2 mm in diameter with 1 cm separation, whereas in the depth electrodes each recording site was 1 mm in diameter with 5 mm separation. A subgaleal electrode at the vertex was used as a reference and the acquired signals were filtered between 0.07 Hz and 200 Hz (half-power boundaries) and sampled at a rate of 500 Hz by an XLTEK EMU128FS system (Natus Medical, Pleasanton, California) or filtered between 0.1 Hz and 256 Hz and sampled at a rate of 512 Hz by a BRAINBOX EEG-1166 system (Braintronics, Almere, Netherlands). Stimulus-triggered electrical pulses were recorded along with the electrophysiological data to allow precise timing of neural responses in relation with the stimuli and the video eye-tracking data.

## Anatomical localization of electrodes

Prior to electrode implantation, patients underwent a T1 weighted 1 mm isometric anatomical MRI scan on a 3-tesla Signa HDx scanner (GE Healthcare, Chicago, Illinois). Following the implant, a thin-slice computed tomography (CT) and a T1-weighted anatomical MRI scan on a 1.5-tesla Signa Excite scanner (GE Healthcare) were acquired in order to aid electrode localization. The pre-implant and post-implant MRI scans were rigidly co-registered using FSL's Flirt (*Jenkinson and Smith, 2001*; *Jenkinson et al., 2002*; RRID:SCR_002823). A similar co-registration was performed on the post-implant MRI and post-implant CT scans. Concatenating these two co-registrations allowed visualizing the post-implant CT scan on top of the pre-operative MR scan while minimizing localization error due to potential brain shift caused by surgery and implantation. Individual contacts were then identified by an expert inspection of the thresholded CT along with the post-op MR and were marked in

each subject's pre-operative MRI native space, aided by BioImage Suite (*Papademetris et al., 2006*; RRID:SCR_002986).

A 3d model of each patient's cortical surface was segmented and reconstructed from the pre-implant MRI using FreeSurfer 5.3 (*Dale et al., 1999*; RRID:SCR_001847). Each electrode contact was snapped to the nearest vertex on the cortical surface (e.g. *Dykstra et al., 2012*). Contacts that were farther away than 10 mm from the surface were excluded from further analyses. In order to allow for integration of observations across subjects, the three-dimensional mesh of each individual cortical surface was standardized by resampling its unfolded spherical form using SUMA (*Argall et al., 2006*; RRID:SCR_005927). This resulted with a cortex-based alignment of each patient to a common template, allowing the visualization of electrodes from different individuals on a single cortical surface while adhering to the electrodes' location in relation with individual gyri and sulci.

## ECoG signal preprocessing

All preprocessing was done using in-house made Matlab (The MathWorks, Inc.) scripts, except for filtering for which EEGLAB's two-way least-squares FIR filtering (*Delorme and Makeig, 2004*; RRID: SCR_007292) was employed. Each channel was downsampled to 500 Hz (if required), re-referenced by the common average across all intracranial channels within the individual patient (following an established practice, see *Liu et al., 2015*) and then notch-filtered to remove 60 Hz line noise. Next, channels were manually inspected for interictal spikes, remaining electromagnetic interferences, eye or eyelid movement artifacts and other noticeable signal contaminations. An additional channel exclusion criterion was the physical position of the electrodes over lesions or ictogenic zones identified by a certificated neurologist. The common average reference was then recomputed using only non-excluded channels and only these channels were considered for further analysis.

Following this initial preprocessing, high-frequency broadband (HFB) response between 70 and 150 Hz was estimated. Eight 10 Hz wide band pass filters were used to split the signal into narrow frequency bands. The momentary amplitude of each band was then estimated by the absolute value of its Hilbert transform. Each of the resulting timecourses was then divided by its mean across time. Next, the eight bands were recombined into a single timecourse by averaging. This procedure is aimed at sampling the relevant frequency range with equal weighting, overcoming the 1/f amplitude spectrum of the LFP signal (*Fisch et al., 2009*; *Ossandón et al., 2011*).

HFB signals were then automatically scanned for widespread outliers by the following procedure: the discrete-time derivative of each channel's HFB was z-normalized and its absolute value was taken. Any timepoint in which the across-channels median of the absolute values z-scores exceeded the value of two was marked for rejection, as well as timepoints distanced up to 200 ms away from it. In the subsequent analyses, any trials including rejected timepoints were excluded.

Finally, the HFB time series of each electrode was downsampled to 250 Hz and normalized again by dividing it by the mean activation during the inter-block gray screen intervals, transforming the time series' units into percent signal change with the inter-block intervals as a 0% change baseline.

## Quantification of visual responses to the object images

The mean HFB activation during 50 ms to 350 ms following the beginning of each trial (marked by the transition from one stimulus exemplar to another) was compared with the mean HFB activation measured during the inter-block baseline periods, in which a gray screen was presented. Only trials that contained no blinks (or gaps, by design) during that interval or in the preceding 100 ms were considered for this analysis (see subsection on blink detection below). Whereas statistical testing against baseline was done using all of the object images, effect-size estimation was based on the optimal (maximally responding) category of each electrode. This response was subtracted by the mean baseline HFB and then divided by the standard deviation of baseline periods, yielding a Glass' Δ effect-size estimate. This estimate was used in the visualization in *Figure 2* and in the effect size criterion for electrode inclusion.

## Region of interest (ROIs) definition

In order to enable averaging across electrodes and patients, the visually responsive electrodes were parceled into seven ROIs (V1,V2,V3,V4,VO, face-selective and high-level non-face selective electrodes, see *Figure 5—figure supplement 1* and *Table 1*) based on functional and anatomical criteria.

First, electrodes functionally qualifying as 'face-selective' (see Results) were assigned to the 'face-selective' ROI. For each of the remaining electrodes, we compared its location on the SUMA standardized template with a surface-based probabilistic topographic atlas, based on a sample of fMRI retinotopic mappings (*Wang et al., 2014*). Each electrode was associated with the topographic map of the maximal probability value over the electrode's snapped vertex, but only if that probability value was at least 25%. When this condition was not met, we searched for qualifying vertices in a 5 mm radius from the snapped vertex. Electrode locations still unlabeled by this procedure were then tested against the V1, V2 (*Fischl et al., 2008*) and MT+ (*Malikovic et al., 2007*) labels derived from post-mortem histology mappings, as provided by FreeSurfer 5.3 (RRID:SCR_001847). Since these labels are cytoarchitectonically defined, they extend further than the effective visual field of the functional retinotopic mapping.

Whereas the atlas by *Wang and colleagues (2014)* describes 25 topographic maps, we required that each potential ROI to be sampled by at least six patients, in order to enable reliable generalization. This resulted with five retinotopic ROIs: V1, V2, V3, V4 and VO (Ventral-Occipital). Dorsal and ventral subdivisions of V1, V2 and V3 were combined. 35 visually responsive electrodes that were associated with maps which were not sufficiently sampled across patients were not assigned to an ROI, hence they were not included in the grand averages and the group mixed-effects analysis, but they are included in the within-electrode analysis (*Figure 5* and *Figure 5—figure supplement 1*).

Finally, the remaining electrodes that were more than 5 mm farther from any of the retinotopic maps or histological labels were grouped together as the 'high-level non-face selective' ROI. All of these electrodes were located anterior to the retinotopic atlas (*Figure 5—figure supplement 1*).

## Time domain deconvolution

In order to isolate the effect of blinks (and gaps) on the neural response, the contribution of the stimuli displayed in the background (e.g. face images) has to be accounted for. A simple approach is to form a template of the response to uninterrupted stimuli and subtract it from the timecourse prior to the segmentation and averaging of blink (or gap) related responses. We opted for a more statistically efficient method that utilizes the entire timecourse without selection (but see *Figure 5—figure supplement 1c*). As often done in the analysis of fast-event related fMRI (*Burock and Dale, 2000*) and recently introduced to the analysis of EEG (*Dandekar et al., 2012*), the timecourse was modeled as a sum of finite impulse response (FIR) basis functions, expressing the contributions of events of different experimental conditions over pre-specified ranges of time lags. By estimating a corresponding linear multiple regression model, the contribution of each experimental condition at each time lag was estimated while taking into account the contributions of all other events.

The structure of the regression design matrix is illustrated in *Figure 3*. The onsets of new face stimuli (i.e. the beginning of face trials) and the onsets of new non-face stimuli (excluding animals) were modeled by two sets of 375 FIR predictors spanning from 0 to 1500 ms post stimulus onset. The onsets of spontaneous blinks, that is, the image disappearances due to spontaneous blinks, were also modeled separately for face and non-face trials: Image disappearances due to spontaneous blinks occurring during were modeled using a set of 187 FIR predictors, spanning from –250 to 500 ms post blink onset, with separate predictors for blinks occurring at face trials and at non-face trials. Each unit pulse within an FIR predictor spanned exactly one timepoint (4 ms). Gaps and voluntary blinks were modeled by four additional predictor sets (again, modeling separately events that happened during face and non-face trials) of the same parameters. In principle, blink and gap offsets (i.e. the reappearance of the stimuli) could have been modeled in parallel with the onsets within the same model. However, we found that such a model suffers from issues related with collinearity. Therefore, the effects of image reappearance were tested by fitting an alternative model in which blink and gap offsets were used instead of onsets. A model incorporating both onsets and offsets simultaneously was used only for the plotting of the stimulus-related responses (*Figure 5b*). Timepoints during animal trials, timepoints rejected due to signal outliers or eye tracking problems and timepoints during the rest breaks were all excluded from this analysis by the inclusion of a dummy nuisance predictor for each excluded timepoint. For conciseness and ease of visualization, the non-selective single electrode responses (*Figure 4a* and *Figure 4—figure supplement 1*) and the group gap-blink response map (*Figure 6*) were derived from a simpler regression model built without the distinction between face and non-face stimuli.

Given an HFB timecourse and a design matrix, the ordinary least square estimator was used to derive regression coefficients: $\beta = (X^T X)^{-1} X^T y$, where $\beta$ is the p-long vector of the estimated regression coefficients, X is the design matrix (n × p) and y is the observed timecourse. p stands for the number of predictors and n stands for the number of timepoints in the observed time course.

The standard errors of these regression coefficients were estimated by an HC3 heteroskedasticity-consistent standard error estimator (*Davidson and Mackinnon, 1993*): $HC3 = (X^T X)^{-1} X^T diag \left[ \frac{e_i^2}{(1-h_{ii})^2} \right] X (X^T X)^{-1}$, where HC3 is the p-long vector of estimated regression coefficients' standard errors, X is the design matrix, $diag[v]$ is a diagonal matrix with vector v on its main diagonal (as in matlab 'diag' function), $e_i$ is the residual at timepoint i and $h_{ii}$ is the 'leverage value' of timepoint i, defined as the i-th entry of the main diagonal of the 'hat' matrix H. H is the matrix that transforms an observed timecourse into a predicted timecourse, defined as $H = X(X^T X)^{-1} X^T$.

The motivation behind using this more involved standard error estimator is that unlike the homoscedastic standard error estimator commonly applied in fMRI GLM analyses, a heteroskedasticity-consistent estimator does not yield error bars of a uniform height across time. Instead, the estimated standard-error reflects the variability of the particular samples that contributed to each timepoint in the deconvolved trace (see *Hayes and Cai, 2007* for introduction and review).

For the testing of single-electrode individual timepoints (*Figure 4* and *Figure 4—figure supplement 1*), the estimated regression coefficients were divided by their standard errors and compared to a standard normal distribution (a large-sample approximation). The resulting p-values were then FDR-corrected in time with a q-value equal to 0.05 divided by the number of visual electrodes within the patient.

## Quantification of activation-dip and reappearance-related overshoot

In each electrode and for each interruption type (gap, spontaneous blinks and voluntary blinks) we quantified the two following components: (a) reappearance-related overshoot, which was measured from the offset (reappearance)-locked deconvolved HFB response and (b) activation-dip, which was measured from the onset (disappearance)-locked deconvolved HFB response. For the reappearance-related overshoot, we measured the integrals (area under curve) below each contiguous cluster of above-zero timepoints following the reappearance and picked the largest one. Similarly, for the activation-dip, we picked the maximal integral above clusters of contiguous below-zero timepoints that followed the stimulus disappearance. The units of both integrals were percent signal change times second.

For the group mixed-effects analysis, the interruptions were randomly split into two halves, each giving rise to a distinct set of predictors. This resulted with two independent deconvolved traces for each interruption type. One trace was used to determine the clusters' temporal extents while the other was used to measure the activation integrals during these intervals. These measurements were averaged across 30 random splits of the data. This procedure guaranteed the response estimates to be non-circular (i.e. free from selection-bias), while preserving the flexibility of tailoring each cortical site with its optimal time windows.

## Within-electrode statistical testing

In order to test the significance of the observed differences between gap-related and blink-related responses at each recording site, we estimated the empirical null-distribution of the difference between them using a random permutation test. Differences in activation-dips and reappearance-related overshoots were tested separately. In each simulation iteration, we shuffled the labels of the single gap-events with the single blink-events (either voluntary or spontaneous blinks, the two blink types were contrasted with gaps in separate tests). This shuffling was restricted such that the counts of gaps and blinks occurring within face trials and non-face trials were kept constant. Then, the deconvolved response traces were re-estimated using the shuffled labels and the difference in activation-dip or reappearance-related overshoot components between gaps and blinks was recorded. The observed differences derived from the original, unshuffled events were assigned with a p-value equal to $p = \frac{b+1}{m+1}$ where b is the number of random permutations with a difference at least as big as the observed difference and m is the number of total simulations (*Phipson and Smyth, 2010*), which

was set to 2000. The resulting p-values were corrected for 2-tailed testing and then were FDR-corrected across all visual responsive electrodes.

In order to test for non-uniformity of the proportions of significant results across ROIs, we used a randomization test of independence (*McDonald, 2009*). This was implemented by measuring the following test statistic: $\chi^2 = \sum_{i=1}^{7} \frac{(O_i - E_i)^2}{E_i}$ where $O_i$ is the observed number of significant electrodes in the i-th ROI and $E_i$ is the expected number of significant electrodes that ROI, assuming uniformity across ROIs (i.e. the number of electrodes in that ROI multiplied by the overall proportion of significant electrodes across all ROIs). The observed statistic was compared with an empirical null-distribution generated by randomly assigning significant electrodes across ROIs while keeping the total number of electrodes within each ROI and the total number of significant electrodes across ROIs fixed.

## Mixed-effects group analysis

Mixed-effects group analysis was implemented using the LME4 package (*Bates et al., 2014*) of the R language (*R Core Team, 2013*). The magnitude of event-related responses (activation-dip or reappearance-related overshoot, each component tested independently) was entered as the dependent variable. The independent variables were interruption type (gap, voluntary blink or spontaneous blink), stimulus category (face or non-face) and ROI (V1, V2, V3, V4, VO, Face-selective or high-level non face-selective). A random intercept model was formulated as response ~ interruption*category*ROI+(1|patient/electrode). Whereas including random slopes when applicable is generally recommended (*Barr et al., 2013*, but see *Bates et al., 2015*), these could not be included for our dataset since they led to over-parameterization (model undefinability). The analyses of duration and latency matched events (*Figure 7—figure supplements 1,2*) used only face-trials; hence their model included only two independent variables, interruption (gap/blink) and ROI. This model was fitted separately for each matching procedure (matching of event onset latency and matching of event duration).

Main effects and interactions were tested using Type III ANOVA with Kenward-Roger approximation for degrees of freedom implemented by the afex R package (*Singmann et al., 2015*). Simple effects were tested within each ROI using lsmeans R package (*Lenth and Hervé, 2015*) and were FDR-corrected for multiple comparisons.

Since initial inspection of the data found greater variability in conditions with greater observed values, we log-transformed all values prior to model fitting in order to better conform to the model's homoscedasticity assumption. This transformation did not qualitatively change the subsequent results. Prior to the log-transform, the data were uniformly shifted in order to avoid negative values and the sign of the activation-dip magnitudes was inverted. For visualization purposes (in *Figure 7*, *Figure 7—figure supplements 1–3*), the estimated coefficients and the locations of their standard error estimates were transformed back to the original scale by the corresponding inverse transforms.

## Blink detection

Patients' blinks were monitored by a video eye tracker (ET) operated monocular at 500 Hz (EyeLink 1000, SR research, Ontario, Canada) and by a single EOG electrode placed above one of the patient's eyebrows (referenced to the ECoG common-average). In order to register the occurrence of a blink, the concurrent presence of EOG and pupil-size (measured by the ET) blink-related artifacts was required. Trials including ambiguous events or missing eye tracking data were excluded from later analysis. In general, both measures picked up reliable blink-related artifacts and were in high agreement. In two of the patients, in which no video-tracking was available (P20 and P25), blinks were detected exclusively by the EOG channel.

## Blink onset and offset timing

As the eyelid gradually covers the pupil, the pupil size estimate of the ET rapidly diminishes, producing a distinctive artifact. Blink onsets (blink-related image disappearances) were registered at the peak acceleration of the measured pupil-size decrease (corresponding with the eyelid passing the pupil's midline). In cases in which tracking was lost before the occurrence of such an acceleration peak, blink onset was marked at the last sample prior to signal loss. Respectively, blink offset (blink-

related image reappearance) was marked by the peak acceleration of the pupil-size increase measured following the reacquisition of the pupil signal.

For the two patients where no eye tracking was available (P20 and P25), electrooculogram (EOG) -based blink timing was used. The EOG signal was filtered (FIR bandpass, 0.2–40 Hz). Remaining slow noise fluctuations were removed by fitting the EOG trace around each blink (±750 ms) with a sum of a fifth degree polynomial baseline and a trapezoid (modeling the blink artifact) with height, onset, sustain and decay times as free parameters. The estimated polynomial baseline was then removed from the trace. Next, the baseline-removed trace was thresholded: the sample that first passed the threshold marked the blink onset and the first sample that crossed back below it marked the blink offset. The individual threshold level for each of the two patients was determined empirically from the other 12 patients that did have concurrent EOG and eye tracking: for each such patient, the median amplitude of the EOG blink-artifact produced by spontaneous blinks was estimated. We scanned the range between 0 and 100% and found the threshold of 22% to best predict the eye-tracker based blink duration from the EOG artifact. Repeating the analyses reported in the results section with EOG-based blink detection and timing for the entire group of 14 patients yielded highly comparable results to those described in the results section above.

### Onset latency matching procedure

The onset matching procedure of blinks with gaps was performed separately for spontaneous and voluntary blinks. The procedure was done within-patient and used only gaps and blinks that occurred during face-trials. Matching was done iteratively: In each matching iteration, one gap and one blink were paired. In order to be eligible to pairing, the two events had to have a latency difference of no more than 50 ms. The decision criterion among multiple possible pairings was the minimization of the difference between the average latency of the paired gaps subset and the average latency of the paired blinks subset. This was repeated until no eligible pairings were available. This procedure resulted with a subset of gaps and a subset of blinks of an equal number of events and of highly similar latency histograms and means (*Figure 7—figure supplement 1a–b*). The data from patients whose events could not be matched (no eligible pairings) were excluded from the matched analysis (P50 for both voluntary and spontaneous blinks, P25 and P32 for spontaneous blinks).

### Duration matching procedure

Alternatively, gaps and blinks were matched for duration in the same manner as they were matched for onset latency. The pairing criterion was no more than 15 ms difference in duration and the choice among multiple pairings was by minimizing the difference between the average duration of the paired gaps subset and the average duration of the paired blinks subset. As in the latency matching, this resulted in equally-sized events subsets with highly similar histograms and means (*Figure 7—figure supplement 2a–b*). Patient P50's events could not be matched using this procedure for both voluntary and spontaneous blinks. Patient P44's spontaneous blinks (but not voluntary blinks) were also excluded.

### False discovery rate (FDR) correction

All FDR corrections applied in the study used 'FDR p-value adjustment' described by *Yekutieli and Benjamini (1999*, eq. 3), implemented by *Winkler (2011)*, ensuring that individual p-values control false discover rate as defined by *Benjamini and Hochberg (1995)*.

## Acknowledgements

We thank the two anonymous reviewers for their constructive comments, Mallorie Lenn for her efficient research assistance and Francesca Strappini for her sound graphic design advice. We are grateful to the participating patients, who contributed their time and effort to this study.

# Additional information

## Funding

| Funder | Grant reference number | Author |
|---|---|---|
| Schweizerischer Nationalfonds zur Förderung der Wissenschaftlichen Forschung | 148388 | Pierre Mégevand |
| National Institutes of Health | EY024776 | Charles E Schroeder |
| United States-Israel Binational Science Foundation | 2013070 | Leon Y Deouell |
| Israel Science Foundation | 1902_14 | Leon Y Deouell |
| Page and Otto Marx Jr. Foundation | | Ashesh D Mehta |
| Helen and Martin Kimmel Award | 7204760501 | Rafael Malach |
| Israel Science Foundation | I-CORE, 7111000508 | Rafael Malach |
| Canadian Institute for Advanced Research | Azrieli Program on Brain, Mind, and Consciousness | Rafael Malach |

The funders had no role in study design, data collection and interpretation, or the decision to submit the work for publication.

## Author contributions

TG, Conception and design, Analysis and interpretation of data, Drafting or revising the article; ID, Conception and design, Acquisition of data, Drafting or revising the article; MM, DMG, PM, EMY, MSG, LM, Acquisition of data, Drafting or revising the article; MH, Implant digital reconstruction; CES, LYD, ADM, RM, Conception and design, Drafting or revising the article

## Author ORCIDs

Tal Golan, http://orcid.org/0000-0002-7940-7473
Pierre Mégevand, http://orcid.org/0000-0002-0427-547X
Leon Y Deouell, http://orcid.org/0000-0002-6147-5208
Ashesh D Mehta, http://orcid.org/0000-0001-7293-1101
Rafael Malach, http://orcid.org/0000-0002-2869-680X

## Ethics

Human subjects: All patients gave fully informed consent, including consent to publish, according to NIH guidelines, as monitored by the institutional review board at the Feinstein Institute for Medical Research, in accordance with the Declaration of Helsinki. Data was obtained as part of protocol number 07-125. Patients had the opportunity to consent prior to electrode implantation and were informed that they may choose to decline or later withdraw from the study without affecting their clinical care. Consent includes agreement to participate with studies of cognitive and sensorimotor processes and publication of any deidentified data obtained. Risks include tedium and potential breach of medical information and are minimized by giving ample breaks and implementation of protocols to deidentify data close to the time of recording. Benefits to the subject include increased monitoring of the electrocorticogram and involvement of research methods to help localize electrodes with respect to preoperative MRI.

# Additional files

## Major datasets

The following previously published dataset was used:

**Database, license,**

| Author(s) | Year | Dataset title | Dataset URL | and accessibility information |
|---|---|---|---|---|
| Wang L, Mruczek RE, Arcaro MJ, Kastner S | 2015 | Probabilistic Maps of Visual Topography in Human Cortex | http://scholar.princeton.edu/napl/resources | Publicly available at Princeton University Website (http://scholar.princeton.edu/napl/resources) |

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
