## [Decision Letter]

Thank you for submitting your article "Human Intracranial recordings link suppressed transients rather than "filling-in" to perceptual continuity across blinks" for consideration by *eLife*. Your article has been reviewed by two peer reviewers, and the evaluation has been overseen by a Reviewing Editor and David Van Essen as the Senior Editor. The reviewers have opted to remain anonymous.

The reviewers have discussed the reviews with one another and the Reviewing Editor has drafted this decision to help you prepare a revised submission.

Summary:

In this paper the authors used ECoG recordings to examine the neural basis of perceptual continuity during an eye blink when visual input is briefly interrupted. They examined responses at earlier and later stages of visual cortical processing in 14 patients during spontaneous blinks, voluntary blinks and during equivalent external gaps in visual stimulation. They found that in early visual cortex responses showed a dip following any interruption in the visual input, irrespective whether it was caused by a spontaneous or voluntary blink or by a gap in visual stimulation. However, at subsequent stages of visual cortical processing, the effect of the blink declined but not that of the gap in visual stimulation. The reduction in the effect of blinks was present in area V4 and was even more pronounced in VO1 and in the ventral temporal cortex. The authors explain the reduction of blink effects at later stages of cortical processing by the operation of the mechanisms that suppress blink-related visual transients, arguing against "filling-in" of the missing stimulus content.

Essential revisions:

Both reviewers found the results interesting and compelling but raised a number of reservations that must be addressed for the paper to be considered for publication in *eLife*. The reservations are both conceptual and methodological.

Conceptual.

The authors should address the point made by Reviewer 1 concerning favoring the "suppression" over "filling-in" and to consider alternative explanations of the reduced blink effect in later visual cortex, such as more sustained responses or memory. Also, the authors should address a possibility that there is a neural signal in another cortical region that distinguishes between blinks and gaps but does not show a visual response.

Data analysis.

1) Please provide justification for the use of deconvolution analysis, rather than trial averaging at the time of post-blink stimulus onset, and respond to the concern about the stability of the deconvolution model.

2) Both reviewers agree that the use of ratios to show differential effect of the gap vs blink (in Figure 6) may inflate small signal differences. They suggest that the authors generate a map of size effects by computing a map of significant differences between responses to gaps and blinks. Please address this point and the suggestion to recompute this comparison of the effects of blinks and gaps.

Technical issues.

1) Reviewer 2 pointed out the presence of fast eye movements during blinks and that there is a possibility that the reported results may affected by "saccadic suppression-like" mechanisms. Please discuss the potential role of eye movement during blinks in the observed results.

2) The same reviewer points out that not all spontaneous blinks cover the pupil completely, ranging from a few hundred microns to 12 mm, and suggests that this may provide the authors with the opportunity to disassociate visual transients from blinks. If possible, analysis involving relating the size of blinks to neuronal activity may provide important insights and will help in interpreting the results.

*Reviewer #1:*

In this paper Golan and colleagues examine with ECoG recordings the effects of blinks on responses in visual cortex. They compare the effects of spontaneous blinks to voluntary as well as to external gaps in which the stimulus is replaced by a gray or black screen. To analyze the effect of the blinks/gaps, they use an innovative analysis in which they sort the trials by the onset of the visual stimulus after the blank/gap. Their results reveal three main findings:

1) in early visual cortex the visual signal shows a dip following the blink/gap and then rebounds. Interestingly, the effect is not different for spontaneous/voluntary blink vs. an external gap. This suggest that early visual cortex shows a decline in response when the retinal stimulus is disrupted and rebounds when the visual stimulus is restored and there is no difference between a blink and an experimenter induced gap in the visual display.

2) Across the hierarchy of retinotopic regions there is a decline in the effect of the blink on response, but there is no difference in the effect of the external gap

3) Starting in an around V4 and more so in VO1 and ventral temporal cortex (VTC) there is a difference in the visual response in the high frequency broadband (HFB) to blinks compared to experimentally generated gaps in the visual display. The authors conclude by suggesting two alternatives of 'filling in' vs. 'blink suppression' to explain the differences in higher order areas between the effects of the blinks and the effects of gaps.

While I find the empirical results of interest to the general field, as understanding the effects of a basic phenomena such as blinking is not well studied, I would urge the authors to exert caution on their theoretical interpretation for the following reasons:

1) Neither 'filling-in' or 'blink suppression' has been operationalized in the manuscript. Therefore, it is difficult to infer from the experimental data that one hypothesis is valid and the other is not.

2) Other mechanisms such as memory and sustained neural response over the blink period should be considered. For example, higher order regions may exhibit sustained responses over blinks but not over the external gaps, or rebound responses after the blink for a new but not an old stimulus. These possibilities do not neatly fill into the theory of 'filling in' or 'blink suppression' but need to be considered.

3) For the theory of blink suppression to be validated there needs to be empirical evidence for the existence of a neural signal, perhaps in regions external to the ones reported here, that differentiates the blink from the gap, but does not show a visual response. Such evidence was not reported in the present manuscript.

Data analyses.

I have several comments regarding the data analyses that require clarification/responses from the authors.

1) Deconvolution: The authors should justify why they use a deconvolution analysis rather than brute force averaging of the trials based on the onset of the stimulus after the blink/gap. Additionally, the duration of events in Figure 3 and the methods is not specified. Further, the events shown in Figure 3 do not seem independent from each other (for example, the duration of one stimulus or one blink includes a series of events that are coded in succession). Thus, I am concerned about the stability of the least squared solution of the deconvolution model based on this coding of events.

2) Figure 6 is interesting as it shows the differential between the effect of the gap vs. the blink across the visual system. This figure presents the data in a very clear way as it shows a map of this differential effect. However, the metric that the authors use is a non-standard one: In the caption they write: 'The circles' face hue is mapped to the ratio between the gap and blink measures and its saturation to their combined magnitude (Pythagorean sum)'. Using ratios especially when signals are small (e.g. in the higher level ROIs) may inflate small signal differences (i.e. the ratio of a division of small numbers may be large). Instead, I suggest to re-plot this figure by generating a map of the effect size (i.e. difference of the responses to gaps vs. blinks) or a map of the t-contrast between the response to gaps vs. blinks. These analyses would be (i) more directly relatable to the prior figures, (ii) will provide more stable summaries of the data, and (iii) will not be prone to the inflation of small signals.

3) For the bar plots in Figure 7, it is unclear what is the time window over which these bars were calculated. While I understand that the dip-related decreases (C, D) may be variable depending on the duration of the blink, the reappearance signal (A, B) is locked to the onset of the visual stimulus following the blink/gap, so a constant time window can be used.

*Reviewer #2:*

This study describes ECoG activity throughout the visual system in response to spontaneous blinks, voluntary blinks, and equivalent visual gaps in order to investigate the neural mechanisms that prevent our perception of visual transients with blinking. The investigators conclude that perceptual stability with blinking results from a suppression of visual transients rather than a 'filling in' of the missing visual information. The authors' data analysis is thorough and very careful. Nevertheless, I have some concerns about potential confounds with the data.

First, the authors focus exclusively on eye lid movements as the critical element in the suppression of visual transients. Blink associated eye movements may also contribute. When looking straight ahead, the eyes rotate nasalward and downward with each blink. Riggs and his colleagues showed that visual suppression occurs during blinking independent of lid closure. Because blink associated eye movements achieve velocities in the range of saccadic eye movements, saccadic suppression-like mechanisms may influence the investigators' results. The authors might consider the role of blink associated eye movements in interpreting their results.

Second, the authors might strengthen their argument about suppression of visual transients with blinking by analyzing ECoG data for blinks that do not cover the pupil or partially cover the pupil. The authors write that 'blink-suppression signals may actively block these detectors of [visual] discontinuity'. Given that spontaneous blink amplitude ranges from a few hundred microns to 12 mm blinks that fully close the eye; the investigators might be able to identify the activity of the blink suppression signals independent of visual transients.

Finally, the authors present a caveat to interpreting the ECoG signals. In Figure 5, the time-locked ECoG response to visual stimuli shows a steady decay across the one second presentation, but the perception of the visual stimulus does not fade during that period. This type of observation raises the question as to how straightforward it is to interpret the meaning of the differences in the ECoG peaks and gaps among the voluntary blinks, spontaneous blinks, and gap stimuli.

---

## [Author Response]

*Essential revisions:*

*Both reviewers found the results interesting and compelling but raised a number of reservations that must be addressed for the paper to be considered for publication in eLife. The reservations are both conceptual and methodological.*

*Conceptual.*

*The authors should address the point made by Reviewer 1 concerning favoring the "suppression" over "filling-in" and to consider alternative explanations of the reduced blink effect in later visual cortex, such as more sustained responses or memory.*

We have revised and extended the Discussion to address these points (see points 1 & 2 below).

*Also, the authors should address a possibility that there is a neural signal in another cortical region that distinguishes between blinks and gaps but does not show a visual response.*

This is now addressed in the revised Discussion (see point 3 below).

*Data analysis.*

*1) Please provide justification for the use of deconvolution analysis, rather than trial averaging at the time of post-blink stimulus onset, and respond to the concern about the stability of the deconvolution model.*

A detailed justification and a fuller explanation of the deconvolution analysis as well as arguments to alleviate the concern of instability are provided in response to Reviewer 1 Data analyses point 1.

*2) Both reviewers agree that the use of ratios to show differential effect of the gap vs blink (in Figure 6) may inflate small signal differences. They suggest that the authors generate a map of size effects by computing a map of significant differences between responses to gaps and blinks. Please address this point and the suggestion to recompute this comparison of the effects of blinks and gaps.*

We have now generated a map of significant differences between gaps and blinks. This replaces the previous versions of Figure 6 and its supplemental figure (see response to Reviewer 1 Data analyses point 2).

*Technical issues.*

*1) Reviewer 2 pointed out the presence of fast eye movements during blinks and that there is a possibility that the reported results may affected by "saccadic suppression-like" mechanisms. Please discuss the potential role of eye movement during blinks in the observed results.*

This is now addressed in the revised discussion (see first response to Reviewer 2).

*2) The same reviewer points out that not all spontaneous blinks cover the pupil completely, ranging from a few hundred microns to 12 mm, and suggests that this may provide the authors with the opportunity to disassociate visual transients from blinks. If possible, analysis involving relating the size of blinks to neuronal activity may provide important insights and will help in interpreting the results.*

Following the reviewer's suggestion, we have tested this possibility thoroughly. Unfortunately, our conclusion is that reliably isolating partial blinks seems to require a different eye tracking system than the one we used during the ECoG recordings (see a detailed discussion in the second response to Reviewer 2).

*Reviewer #1:*

*While I find the empirical results of interest to the general field, as understanding the effects of a basic phenomena such as blinking is not well studied, I would urge the authors to exert caution on their theoretical interpretation for the following reasons:*

*1) Neither 'filling-in' or 'blink suppression' has been operationalized in the manuscript. Therefore, it is difficult to infer from the experimental data that one hypothesis is valid and the other is not.*

We thank the reviewer for this call for clarification.

Regarding 'filling-in' – we have added the following explicit operationalization to the Introduction:

"Operationally, a filling-in mechanism would be reflected in continuous neural activity across blinks but not across gaps despite the decrease in retinal input common to both."

This prediction, tested by contrasting the HFB dips that followed gaps with the HFB dips that followed blinks, was not fulfilled.

The observed effect in high-level visual cortex, greater gap-related transients compared with blink-related transients, is unrelated with the filling-in hypothesis since the latter relates to the ongoing, continuous activity. In order to interpret the reduced blink-related transient, we introduced the possibility of extra-retinal suppression of blink-related neural responses. This is now clarified in the revised Discussion:

"Given the evidence from early visual cortex of a similar retinal impact of blinks and gaps, the most likely (but not exclusive, see below) interpretation of the reduction in blink-related transients in higher-level visual cortex is extra-retinal suppression of the cortical neural response to the retinal impact of blinks."

Since we observed reduced responses and not actual IPSPs, there are also potential alternative interpretations. The revised Discussion was expanded (see point 2, below) to address this possibility.

Please note that the term 'blink suppression' has three different meanings in the literature: 1. the suppression of the perception of blinks, as measured by report, 2. the suppression of neural responses to blinks, as measured by electrophysiology and 3. prevention of blinks from being executed. We therefore slightly edited the wording in the Introduction (third paragraph) to move away from this ambiguity.

*2) Other mechanisms such as memory and sustained neural response over the blink period should be considered. For example, higher order regions may exhibit sustained responses over blinks but not over the external gaps, or rebound responses after the blink for a new but not an old stimulus. These possibilities do not neatly fill into the theory of 'filling in' or 'blink suppression' but need to be considered.*

We hope that under our operational definition of 'filling-in' in the revised Introduction (see above) it is now clearly conveyed that sustained HFB responses over blinks but not over external gaps are precisely the operational predication made by the filling-in hypothesis in our study. As to suppression of transient blink-related responses, we again agree with the reviewer that this mechanism should be considered tentative at this point and this is now pointed out in the text (subsection “High-level visual cortex shows differential responses to perceived vs. unperceived perturbations”, third paragraph). In particular, we agree that the potential role of adaptation (implicated from reviewer's comparison of new vs. old stimuli) is intriguing. It is now discussed explicitly as an alternative mechanism to blink-related suppression of transients:

"An additional potential factor that may be involved in the observed reduction of blink-related transients is neural adaptation. Consider the following scenario, not tested by the current experiment, in which the presented stimulus is swapped during the blink. Would the new stimulus induce a reappearance-related overshoot as in gaps? A positive answer to this question would indicate that gaps (but not blinks) cause a release from adaptation, offering a rather different (yet extra-retinal) explanation of the high-level differential response to gaps and blinks."

The potential role of memory is now also mentioned in the revised Discussion:

"Last, it is worth considering that the perceived continuity across blinks may rely on aspects of neural activity that were not probed by the current recordings. For example, while HFB's ignition-like dynamics are consistent with the timescale of iconic memory, ongoing maintenance of information across larger timescales (as in short-term memory) is not correlated with HFB increases (e.g. Noy et al., 2015)"

*3) For the theory of blink suppression to be validated there needs to be empirical evidence for the existence of a neural signal, perhaps in regions external to the ones reported here, that differentiates the blink from the gap, but does not show a visual response. Such evidence was not reported in the present manuscript.*

We thank the reviewer for bringing this up. We indeed did not observe such a signal. This is now explicitly stated in the revised Discussion:

"The potential source of the suppressing signal was not observed in the current study: neural activity that reliably preceded the execution of blinks was not detected in any of our recording sites. This might be related either to our incomplete coverage of oculomotor cortical regions or to the potential sub-cortical locus of the (spontaneous) blink generator (Kaminer et al., 2011), which was obviously not covered by our recordings."

*Data analyses.*

*I have several comments regarding the data analyses that require clarification/responses from the authors.*

*1) Deconvolution: The authors should justify why they use a deconvolution analysis rather than brute force averaging of the trials based on the onset of the stimulus after the blink/gap.*

In most electrophysiological experimental paradigms, the stimuli are presented with considerable inter-stimulus intervals, assuring that each timepoint registers a neural response triggered by a single event. The standard, brute-force event-related averaging is well justified only in this scenario; once neural responses overlap, event-related averaging yields wrong response estimates since it implicitly assumes that there is no overlap (see discussion of overlap correction in Smith and Kutas, 2015).

In the current manuscript, we study the modulation caused by blinks (and gaps) to the HFB response related with continuously presented stimuli, hence neural response overlap is unavoidable. Since the stimulus-related responses are far from being stationary (i.e. they change throughout the trial, as can be seen in Figure 5), ignoring this overlap would produce erroneous results. We prepared an additional supplemental figure to illustrate this problem by the example of gap-related response estimation in a single electrode (Figure 3—figure supplement 1).

As we show below (see our third response to Reviewer 1 Data analyses point 1), the heuristic approach of removing a template of uninterrupted trials prior to blink (or gap)–related averaging yields estimates similar to those produced by the more principled deconvolution approach, which explicitly models the overlap between all experimental events. The main reason for our preference of the latter over the former is that the deconvolution approach exploits all available timepoints, including those in interrupted trials, in estimating the stimulus-related response.

*Additionally, the duration of events in Figure 3 and the methods is not specified.*

We now understand that this figure's legend was not explicit enough. Due to visualization constraints, we are showing in Figure 3 schematic model with a few very wide pulses (100 ms wide), whereas the actual analysis used many narrow pulses (4 ms pulses, to match the 250 Hz sampling rate we have used).

We have added a clarification in the figure legend:

"Each response is composed of a sequence of non-overlapping unit pulses (4ms-wide pulses were used for the actual 250 Hz HFB timecourse, here a less detailed, 10 Hz model is presented)".

This is now emphasized again in the Methods:

"Each unit pulse within an FIR predictor spanned exactly one timepoint (4ms)."

*Further, the events shown in Figure 3 do not seem independent from each other (for example, the duration of one stimulus or one blink includes a series of events that are coded in succession). Thus, I am concerned about the stability of the least squared solution of the deconvolution model based on this coding of events.*

We believe that this concern is related to the lack of clarity of Figure 3, now addressed in the revised legend. Each experimental condition (e.g. 'face-images', 'blinks over face-images', 'gaps over non-face images' and so on) was modeled by a set of non-overlapping pulses, each pulse spanning exactly one timepoint. Hence, for a particular experimental event (e.g. a particular blink), these predictors are completely independent from each other, as they form a standard (and hence orthogonal) basis (e.g. [1 0 0],[0 1 0],[0 0 1]). This reflects the fact that when there is no overlap between the responses to different experimental events, the deconvolution analysis is equivalent with standard event-related averaging (see Dandekar et al., 2012). Hence the threat of non-independance can come only from systematically overlapping events. Since gaps and blinks did not appear in all of the trials and when they appeared, they had highly variable onset latencies (gaps by design, blinks empirically, see Figure 1—figure supplement 1), the potential for a collinearity issue in this model is very low.

Furthermore, empirically, a direct comparison of the deconvolution derived grand-averages (Figure 5) and the grand-averages derived by standard averaging preceded by the subtraction of a response template estimated from trials with no gaps and blinks (Figure 5—figure supplement 1, panel C) showed a very good correspondence. This correspondence constitutes a strong indicator that the effects we describe are not an artifact of the deconvolution approach or a statistical fluke due to model instability.

*2) Figure 6 is interesting as it shows the differential between the effect of the gap vs. the blink across the visual system. This figure presents the data in a very clear way as it shows a map of this differential effect. However, the metric that the authors use is a non-standard one: In the caption they write: 'The circles' face hue is mapped to the ratio between the gap and blink measures and its saturation to their combined magnitude (Pythagorean sum)'. Using ratios especially when signals are small (e.g. in the higher level ROIs) may inflate small signal differences (i.e. the ratio of a division of small numbers may be large). Instead, I suggest to re-plot this figure by generating a map of the effect size (i.e. difference of the responses to gaps vs. blinks) or a map of the t-contrast between the response to gaps vs. blinks. These analyses would be (i) more directly relatable to the prior figures, (ii) will provide more stable summaries of the data, and (iii) will not be prone to the inflation of small signals.*

Following this suggestion, we have sought to produce a more standard, t-contrast-like map. Since the overshoots are not estimated at the single-trial level (we first apply the deconvolution model on the entire time of each electrode and then detect the overshoot timing from the deconvolved response), the comparison between gaps and blinks at the electrode level is based on permutation testing, not on a two-sample t-test. Hence, instead of t-statistics, we used the minus of the common logarithm of the p-value, i.e. -log10(p). This translates p=0.01 into 2, p=0.001 into 3 and so on. The revised Figure 6 (and its supplement) uses this quantity as its color scale.

In order to keep the color scale easy to understand, we used p-values and not -log10(pvalues) for the tick labels, but this is just a matter of notation.

We have considered other alternatives but they are less suitable for this case; The same limitations related to the fact that this is an electrode-level analysis and not a single-trial analysis exclude the usage of normalized effect-size measures such as Cohen's d. Using non-standardized effect size, that is plotting the raw difference between gaps and blinks regardless of their magnitudes, is technically feasible but clearly improper since different electrodes show different overall response amplitudes.

*3) For the bar plots in Figure 7, it is unclear what is the time window over which these bars were calculated. While I understand that the dip-related decreases (C, D) may be variable depending on the duration of the blink, the reappearance signal (A, B) is locked to the onset of the visual stimulus following the blink/gap, so a constant time window can be used.*

The main motivation for using a variable time window also for the reappearance signal is the different response latencies in different visual sites. We now further clarify this point as follows:

"This procedure directly addressed the response-latency variability across electrodes, reflected in the travelling wave-like nature of the responses across the visual hierarchy (see Video 1)."

The traveling wave-like nature of the reappearance-related overshoot can be seen also in the grand-averages. This wave appears first in early visual cortex and then gradually proceeds forward. Therefore, an unbiased estimation of the reappearance-related overshoot, without contaminating it with the dip, would require setting a time window for each recording site independently. Since unfavorable selection of this window might create spurious results by inadvertently offsetting dips and overshoots, the option of a variable window was chosen.

*Reviewer #2:*

*This study describes ECoG activity throughout the visual system in response to spontaneous blinks, voluntary blinks, and equivalent visual gaps in order to investigate the neural mechanisms that prevent our perception of visual transients with blinking. The investigators conclude that perceptual stability with blinking results from a suppression of visual transients rather than a 'filling in' of the missing visual information. The authors' data analysis is thorough and very careful. Nevertheless, I have some concerns about potential confounds with the data.*

*First, the authors focus exclusively on eye lid movements as the critical element in the suppression of visual transients. Blink associated eye movements may also contribute. When looking straight ahead, the eyes rotate nasalward and downward with each blink. Riggs and his colleagues showed that visual suppression occurs during blinking independent of lid closure. Because blink associated eye movements achieve velocities in the range of saccadic eye movements, saccadic suppression-like mechanisms may influence the investigators' results. The authors might consider the role of blink associated eye movements in interpreting their results.*

We thank Reviewer 2 for raising this possibility. Following the comment we have now considered this in the revised Discussion:

"How does the visual cortex receive the information that discriminates between gaps and blinks? […] Alternatively, given the dramatic retinal impact of eyelid closure, it seems plausible that at least one source of this motor-sensory pathway should be blink-specific, hypothetically originating at the facial motor nucleolus, where the lower motor neurons controlling eyelid closure are located."

*Second, the authors might strengthen their argument about suppression of visual transients with blinking by analyzing ECoG data for blinks that do not cover the pupil or partially cover the pupil. The authors write that 'blink-suppression signals may actively block these detectors of [visual] discontinuity'. Given that spontaneous blink amplitude ranges from a few hundred microns to 12 mm blinks that fully close the eye; the investigators might be able to identify the activity of the blink suppression signals independent of visual transients.*

We thank the reviewer for bringing this interesting opportunity to our attention. Unfortunately, our instrumentation, in conjunction with the blinking-pattern of our subjects, did not allow us to isolate enough retinally-ineffective partial blinks to enable the suggested analysis.

The video eye-tracker we have used (EyeLink 1000, a widely used system in vision research) does not track the eyelids- but rather the change in pupillary area. Thus, the most relevant partial blinks (i.e. blinks that were too small to cross the top edge of the pupil) could not be detected by the eye tracker. We did attempt to compare the responses for short vs. long duration blinks as an indirect estimate for the contrast of larger partial blinks with complete blinks. However, we did not find reliable duration-dependent differences in the blinks' onset response. This might be related to the fact that even the shortest blinks we detected still covered a considerable part of the pupil.

*Finally, the authors present a caveat to interpreting the ECoG signals. In Figure 5, the time-locked ECoG response to visual stimuli shows a steady decay across the one second presentation, but the perception of the visual stimulus does not fade during that period. This type of observation raises the question as to how straightforward it is to interpret the meaning of the differences in the ECoG peaks and gaps among the voluntary blinks, spontaneous blinks, and gap stimuli.*

We fully agree with this comment. Following the reviewers' comment we have now expanded on this issue a bit further as follows:

"Additional evidence for dissociation between momentary activity reductions and perceptual disappearance in high-order visual areas can be found in the consistent and rapid decline we observed in the HFB activity in these regions during the one-second long display of the stimulus (Figure 5), an effect with no apparent perceptual correlate[…] It is tempting to speculate based on these converging phenomena that perhaps the perceptually-relevant signals, at least in high-order visual areas, are phasic bursts rather continuous activations."